# Improving the Effective Receptive Field of Message-Passing Neural Networks

**Shahaf E. Finder** [1 2]   **Ron Shapira Weber** [1 2]   **Moshe Eliasof** [3]   **Oren Freifeld** [1 2 4]   **Eran Treister** [1 2]

## Abstract

Message-Passing Neural Networks (MPNNs) have become a cornerstone for processing and analyzing graph-structured data. However, their effectiveness is often hindered by phenomena such as over-squashing, where long-range dependencies or interactions are inadequately captured and expressed in the MPNN output. This limitation mirrors the challenges of the Effective Receptive Field (ERF) in Convolutional Neural Networks (CNNs), where the theoretical receptive field is underutilized in practice. In this work, we show and theoretically explain the limited ERF problem in MPNNs. Furthermore, inspired by recent advances in ERF augmentation for CNNs, we propose an Interleaved Multiscale Message-Passing Neural Networks (IM-MPNN) architecture to address these problems in MPNNs. Our method incorporates a hierarchical coarsening of the graph, enabling message-passing across multiscale representations and facilitating long-range interactions without excessive depth or parameterization. Through extensive evaluations on benchmarks such as the Long-Range Graph Benchmark (LRGB), we demonstrate substantial improvements over baseline MPNNs in capturing long-range dependencies while maintaining computational efficiency.

## 1. Introduction

Graph Neural Networks (GNNs) have emerged as powerful tools for solving a wide range of problems modeled as graphs. Applications span diverse domains, including combinatorial optimization (Cappart et al., 2023), particle

[1]Department of Computer Science, Ben-Gurion University, Israel [2]Data Science Research Center, Ben-Gurion University, Israel [3]Department of Applied Mathematics and Theoretical Physics, University of Cambridge, United Kingdom [4]School of Brain Sciences and Cognition, Ben-Gurion University, Israel. Correspondence to: Shahaf E. Finder <finders@post.bgu.ac.il>.

*Proceedings of the 42nd International Conference on Machine Learning*, Vancouver, Canada. PMLR 267, 2025. Copyright 2025 by the author(s).

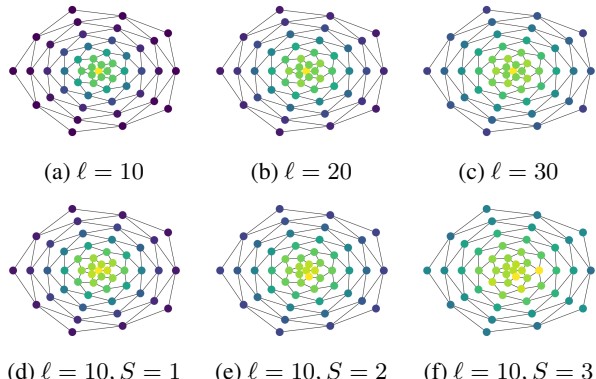

(a) $\ell = 10$     (b) $\ell = 20$     (c) $\ell = 30$

(d) $\ell = 10, S = 1$   (e) $\ell = 10, S = 2$   (f) $\ell = 10, S = 3$

Figure 1: Measuring the contribution of each node to the output of the central node in a graph with a maximal distance of 10 hops from the center. A brighter color marks a larger contribution. We can see a decay akin to Luo et al. (2016). (a)-(c) are MPNNs (GCN) with 10, 20, and 30 MP layers, while (d)-(f) are IM-MPNNs with a GCN backbone with 10 MP layers and 1, 2, and 3 scales.

physics (Shlomi et al., 2020), and social network analysis (Fan et al., 2019). Among the GNN frameworks, message-passing Neural Networks (MPNNs) are particularly prevalent. The core idea of MPNNs lies in local message-passing, where nodes aggregate features from their immediate neighbors, followed by an aggregation operation that updates node representations layer by layer. However, the effectiveness of MPNNs is often hindered by the phenomenon of over-squashing (Alon & Yahav, 2021), where information from distant nodes is ineffectively aggregated, leading to limited expressiveness for long-range interactions.

While over-squashing was initially attributed to bottleneck edges (Alon & Yahav, 2021), more recent analyses emphasize the role of inter-node distances (Di Giovanni et al., 2023). This perspective aligns closely with challenges observed in Convolutional Neural Networks (CNNs), where the Effective Receptive Field (ERF) is often smaller than the theoretical receptive field (Luo et al., 2016). In this work, we further analyze this phenomenon and show that the contribution of node $v$ to the output of node $u$ decays exponentially by the travel distance between them (see Figure 1).

In CNNs, the ERF issue can be mitigated by increasing the

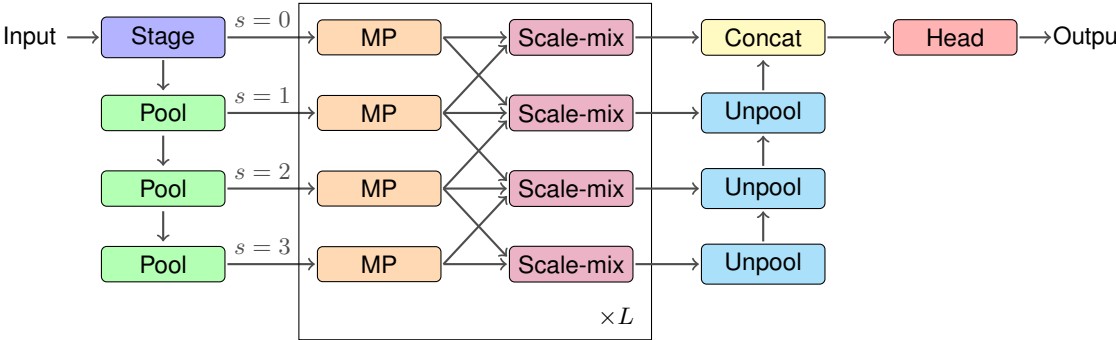

Figure 2: IM-MPNN architecture for scales=3. The input is first passed through an encoding stage (PE, SE, Graph features, etc.). Then, the graph is coarsened $S$ times. MP protocols (GCN, GINE, GatedGCN, etc.) are performed on the $S + 1$ scales of the graph separately. Scale-mix layers pass the information between consecutive scales, matching each graph node with its parent and child from the coarsening process. The process is repeated $L$ times. The coarse graphs are unpooled, and the node features are concatenated to the parent node in the original graph. A GNN head is used according to the task.

kernel size (Ding et al., 2022). However, adapting such approaches to GNNs is nontrivial due to the irregular structure and complexity of the graph's node connectivity. In recent work, Finder et al. (2024) proposed an alternative approach to increase the ERF of CNNs. That method involves small kernel convolutions over multiple scales of the image.

We draw inspiration from such multiscale methodologies in CNNs and propose an Interleaved Multiscale Message-Passing Neural Networks (IM-MPNN) architecture to address the limitations of limited effective receptive fields in GNNs. Our approach introduces hierarchical coarsening of the graph, each reduces the graph's resolution while preserving critical structural information. Message-passing operations are performed at each scale, enabling the model to capture long-range dependencies through fewer layers. To further enhance information flow, we implement inter-scale mixing operations, where nodes receive aggregated features from both higher- and lower-resolution representations. Finally, the features from all scales are combined and projected back to the original graph for downstream tasks. Our architecture is given in Figure 2, and is detailed in Section 4.

**Our Main Contributions** are as follows:

- We formalize the concept of effective receptive fields for MPNNs, drawing parallels to similar challenges in CNNs.

- We propose IM-MPNN – an architecture that leverages hierarchical coarsening and scale mixing to expand the receptive field of GNNs without increasing computational complexity significantly.

- Through experiments on benchmarks such as the Long-Range Graph Benchmark (LRGB), we demonstrate the superior performance of IM-MPNN in capturing long-range dependencies and mitigating over-squashing.

By integrating multiscale representations with message-passing, IM-MPNN offers a principled way to address fundamental limitations of traditional GNN architectures, paving the way for more expressive and scalable graph models. Our code is available at https://github.com/BGU-CS-VIL/IM-MPNN

## 2. Related Work

In this section, we overview several topics related to our work. Specifically, we discuss the over-squashing phenomenon in MPNNs and its connection to their receptive field, as well as review hierarchical MPNN models.

**Over-squashing in MPNNs.** Over-squashing in MPNNs, which hampers information transfer across distant nodes (Alon & Yahav, 2021), has prompted various mitigation strategies. *Graph rewiring* methods like SDRF (Topping et al., 2022) densify graphs as a preprocessing step, while approaches such as GRAND (Chamberlain et al., 2021b), BLEND (Chamberlain et al., 2021a), DRew (Gutteridge et al., 2023), and aAsyn (Chen et al., 2024) dynamically adjust connectivity based on node features. Transformer-based models (Dwivedi & Bresson, 2021; Rampášek et al., 2022) bypass over-squashing with all-to-all message-passing. Another direction uses *non-local dynamics* to enable dense communication, as in FLODE (Maskey et al., 2023), which leverages fractional graph shifts, QDC (Markovich, 2024) with quantum diffusion kernels, and G2TN (Toth et al., 2022), which captures diffusion paths, or weight space constraints (Gravina et al., 2024). While effective in mitigating over-squashing, these methods often increase computational complexity due to dense propagation operators. Moreover, they are often limited by their reliance on the original graph resolution, i.e., its number of nodes and edges.

**The Receptive Field of MPNNs.** We argue that the relation to the receptive field can be mitigated by looking at a coarser version of the graph. E.g., when considering a single hop on the 2nd coarsened scale of a graph, it is related to a receptive field of about four hops. However, all the nodes along the way were used only once for the aggregation function, and therefore, the "exponential" increase in information is mitigated along this path. An earlier study by Nicolicioiu & Duta (2019) focuses on analyzing the ERF in GCNs and Self-Attention layers and showing it on synthetic examples. In contrast, our work addresses the limitations of the ERF in GCNs by introducing IM-MPNN, a hierarchical multiscale framework that explicitly mitigates over-squashing and enhances long-range dependency modeling. Furthermore, we extend the evaluation to diverse benchmarks like LRGB, demonstrating both theoretical and practical improvements.

GeniePath (Liu et al., 2019) introduces an adaptive path layer comprising two complementary functions: one for breadth exploration, which learns the importance of neighborhoods of varying sizes, and another for depth exploration, which filters signals aggregated from neighbors at different hops. This design allows the model to adaptively select the most relevant receptive field for each node, enhancing its ability to capture complex dependencies. Similarly, Ma et al. (2021) propose a structural adaptive receptive field mechanism that enables the network to adjust its receptive field in response to the underlying graph topology. By learning the optimal receptive field for each node, the model effectively balances the trade-off between capturing local and global information, thereby improving performance on tasks requiring an understanding of both. Ma et al. (2023) extend this idea by learning discrete adaptive receptive fields for GCNs, further enhancing their ability to model heterogeneous graphs with varying local structures.

**Hierarchical MPNNs.** Hierarchical approaches to GNNs aim to leverage multiscale representations of graph data to enhance the modeling of both local and global structures. Gao & Ji (2019) developed the Graph U-Net architecture, which employs hierarchical pooling and unpooling operations to capture multiscale features while preserving structural information, making it well-suited for tasks requiring both fine-grained and high-level graph representations (e.g., Yang et al., 2024). Collectively, these hierarchical methods provide a robust foundation for addressing the limitations of traditional GNNs in terms of scalability and expressiveness. Zhong et al. (2023) proposed a framework for hierarchical GNNs that combines localized node feature aggregation with hierarchical graph pooling to better handle large-scale and complex graphs. Similarly, Vonessen et al. (2024) introduced the concept of hierarchical support graphs, which builds a layered representation of the graph to facilitate

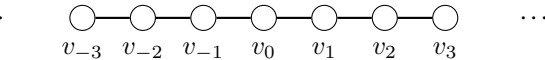

Figure 3: An infinitely-long linear graph.

efficient and scalable message-passing, and Eliasof et al. (2023a) develop a wavelet based multiscale approach for the compression of node features in GNNs. Lastly, in the context of beyond MPNNs, Luo (2023) and Zhang et al. (2022) introduced a hierarchical encoding mechanism for graph transformers, effectively capturing multi-scale graph structure using hierarchical distances to improve the expressive power of transformers.

## 3. Message-Passing Effective Receptive Field

In this section we formulate the effective receptive field of MPNNs. In particular, we draw inspiration from the analysis provided by Luo et al. (2016), where a CNN with $n$ convolutional layer with kernel size $k \times k$ is considered, and it is shown that for $k \geq 2$ the contribution of each pixel in the theoretical receptive field of a certain output decays as a squared exponential with respect to its distance from the output unit. Here, we provide evidence of a similar effect on graphs processed with MPNNs.

**Notations.** Throughout this paper we consider a graph $\mathcal{G} = (\mathcal{V}, \mathcal{E})$ where $\mathcal{V}$ is a set of $n$ nodes, and $\mathcal{E} \subseteq \mathcal{V} \times \mathcal{V}$ is a set of $m$ edges. Additionally, the graph can be described using its adjacency matrix $\mathbf{A} \in \mathbb{R}^{n \times n}$, which is a binary matrix whose $(i, j)$-th entry is one if $(i, j) \in \mathcal{E}$, and zero otherwise. We denote the graph Laplacian as $\mathbf{L} = \mathbf{D} - \mathbf{A}$, where $\mathbf{D}$ is the graph degree matrix. The $i$-th node is associated with a hidden feature vector $\mathbf{x}_i^{(\ell)} \in \mathbb{R}^c$ with $c$ features, which provides a representation of the node at the $\ell$-th hidden layer of the network. The term $\mathbf{x}^{(\ell)} = [\mathbf{x}_0^{(\ell)}, \ldots, \mathbf{x}_{n-1}^{(\ell)}]^\top$ is an $n \times c$ matrix that represents the nodes features of the $\ell$-th hidden layer.

### 3.1. Linear Graph Analysis

We start by analyzing the case of a linear, sequence-like graph without self-loops, as illustrated in Figure 3. Without loss of generality, we theoretically analyze the contribution of distant nodes to the central node $v_0$, center after applying a stack of graph convolutions with uniform weights.

Formally, given an infinitely-long linear graph with nodes $\mathcal{V} = \{\ldots, v_{-3}, v_{-2}, v_{-1}, v_0, v_1, v_2, v_3, \ldots\}$, and examine the feature value $\mathbf{x}_0^\ell$ of node $v_0$ after $\ell$ convolution applications. At $\ell = 0$, the feature of $v_0$ is initialized as:

$$\mathbf{x}_0^0 = v_0. \tag{1}$$

At $\ell = 1$, the feature of $v_0$ is influenced by its neighbors:

$$\mathbf{x}_0^1 = v_{-1} + v_1. \qquad (2)$$

At $\ell = 2$ and $\ell = 3$, the feature of $v_0$ depends on contributions propagated through its neighbors:

$$\mathbf{x}_0^2 = \mathbf{x}_{-1}^1 + \mathbf{x}_1^1 = v_{-2} + 2v_0 + v_2. \qquad (3)$$
$$\mathbf{x}_0^3 = \mathbf{x}_{-1}^2 + \mathbf{x}_1^2 = v_{-3} + 3v_{-1} + 3v_1 + v_3. \qquad (4)$$

As the process continues, the contribution of all nodes to the feature $\mathbf{x}_0^\ell$ after $\ell$ convolutions follows Pascal's triangle:

$$\mathbf{x}_0^\ell = \sum_{i=0}^{\ell} \binom{\ell}{i} v_{2i-\ell}, \qquad (5)$$

where $\binom{\ell}{i}$ are the binomial coefficients.

To analyze the relative contribution of nodes, we normalize the coefficients by $\frac{1}{2^\ell}$, so that the feature $\mathbf{x}$ corresponds to a probabilistic distribution. In particular, this yields the binomial distribution:

$$X \sim B\left(\ell, \tfrac{1}{2}\right), \qquad (6)$$

where the probability mass function of $X$ describes the distribution of contributions to any node from node $v_0$.

Using this binomial distribution, the cumulative contribution of the left-most $k$ nodes from $v_0$ (i.e., $v_{-\ell}, ..., v_{-\ell+k-1}$) is:

$$P(X \le k) = \sum_{i=0}^{k} \binom{\ell}{i} \frac{1}{2^\ell}. \qquad (7)$$

Using Hoeffding's inequality, we get the upper bound:

$$P(X \le k) \le \exp\left(-2\left(\tfrac{1}{2} - \tfrac{k}{\ell}\right)^2 \ell\right). \qquad (8)$$

That is, the relative contribution of nodes to the feature of $v_0$ at the limit decreases exponentially with their distance from $v_0$. This decay is quantified by the binomial distribution's tail bound, indicating that distant nodes have exponentially smaller influence.

### 3.2. Characterizing the ERF on Graphs

Building on the linear graph case presented above, we now provide a qualitative analysis of the ERF in diffusion processes on graphs, which are the core mechanism of many GNN architectures that mimic the behavior of the diffusion ordinary differential equation (ODE), as studied by Nt & Maehara (2019); Chamberlain et al. (2021b); Eliasof et al. (2023b); Choi et al. (2023) and others, as thoroughly reviewed by Han et al. (2024). Such methods rely on the fact that diffusion-based GNNs can be thought of as a discretization of the diffusion ODE in time

$$\frac{\partial \mathbf{x}}{\partial t} = -\mathbf{L}\mathbf{x}, \quad \mathbf{x}(t=0) = \mathbf{x}_0, \quad t \in [0, T], \qquad (9)$$

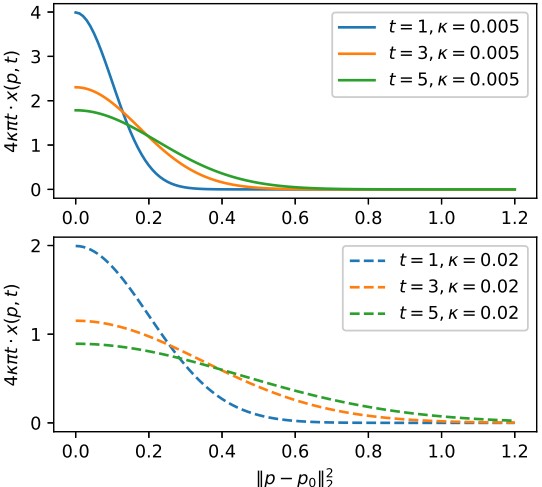

Figure 4: The spread of a point source in time according to Equation (12), for $d = 2$. For $\kappa = 0.005$, at time $t = 1$ the ERF is about 0.3, while for $t = 5$, it grew to about 0.6 only. The ERF is more spread for a higher value of $\kappa$.

where $\mathbf{L}$ is the graph Laplacian, $\mathbf{x}$ are node features that evolve through time (the equivalent of layers), $\mathbf{x}_0$ are the initial node features, and the integration time $T$ typically corresponds to the number of layers used in the network.

Specifically, the interpretation proposed in models that rely on Equation (9), assumes that the graph Laplacian $\mathbf{L}$ discretizes the continuous negative Laplacian $-\Delta$, under some geometry in dimension $d$, i.e., node coordinates $\mathbf{p} \in \mathbb{R}^{n \times d}$, up to a constant denoted by $\kappa$. The coordinates $\mathbf{p}$ can be induced from a given graph Laplacian, e.g., by considering the Laplacian eigenvectors (Belkin & Niyogi, 2001).

Then, Equation (9) is a discretization of the continuous partial differential equation (PDE)

$$\frac{\partial x(t)}{\partial t} = \kappa \Delta x(t), \quad x(t=0) = \mathbf{x}_0, \quad t \in [0, T], \qquad (10)$$

where $x(t)$ is a continuous feature vector in time $t$ with associated coordinates function $p$ (i.e., $\mathbf{x}$ is a discrete version of the continuous map $x$ on the coordinates $\mathbf{p}$).

Similarly to the discussion in Section 3.1, we focus on the distribution of node features in time and space when the initial condition is given by a point source located at around $p_0 \in \mathbb{R}^d$, while all other coordinates have an initial feature of zero. Explicitly, this initial condition is given by:

$$x(t=0) = \mathbf{x}_0 = \delta(p_0), \qquad (11)$$

where $\delta(\cdot)$ is Dirac's function. In this case, the solution of

Equation (10) in an infinite domain $\mathbb{R}^d$ (Pattle, 1959) is

$$x(p,t) = \frac{1}{(4\kappa\pi t)^{d/2}} \exp\left(-\frac{\|p - p_0\|_2^2}{4\kappa t}\right). \qquad (12)$$

From Equation (12), we gain two key insights: (i) the co-efficient multiplying the exponential term in Equation (12) shows that the signal decays in time, i.e., the node features norm decays over time; and (ii) the spatial behavior around the source point $p_0$ decays exponentially. That is, we see that regardless of the dimension $d$, the influence of the source point on other points in space decays exponentially as we move away from the source $p_0$. Furthermore, we note that, as $t$ grows in the denominator inside the exponential term, which is equivalent to a deeper network since layer number $\ell$ takes the role of time $t$, features continue to spread, but at a slow rate. In other words, the typical ERF in standard MPNNs is limited and behaves according to Equation (12). In Figure 4, we illustrate this behavior.

## 4. Interleaved Multiscale Message-Passing

Following our analyses in Section 3 on the limited ERF of MPNNs, we introduce IM-MPNN—an approach to improve ERF through the lens of *interleaved multiscale* MPNNs. We start by discussing our design principles, motivated by the understandings from Section 3, followed by a description of our method, and a complexity analysis.

### 4.1. Design Principles

In this section we outline the main design choices on which we build our IM-MPNN, which we present in Section 4. Specifically, we discuss: (i) the contribution of multiscale feature processing with MPNNs; and (ii) the interleaving of different scales.

**Multiscale Processing.** The analysis in Section 3.2 showed that MPNNs suffer from an exponentially-decaying ERF. We propose to alleviate this problem by considering coarser representations of the input graph, obtained via graph pooling such as Graclus (Dhillon et al., 2007, more information in Appendix A). Different types of pooling procedures can also be considered (e.g., Ying et al., 2018).

We note that when considering different graph Laplacians $\mathbf{L}_{n_1}, \mathbf{L}_{n_2}, \ldots, \mathbf{L}_{n_k}$ of different scales with resolutions (number of nodes) $n_1 > n_2 > \ldots > n_k$, we obtain equivalent representations of the continuous Laplacian (up to discretization differences), each with a different constant $\kappa$. To understand this equivalence, let us consider a 2D mesh grid, where each node is connected to its four neighbors, and whose grid step-size is $h$, which determines the grid resolution, i.e., the number of nodes. In this case, the graph

Laplacian is expressed by the kernel

$$\hat{\mathbf{L}} = \begin{bmatrix} 0 & -1 & 0 \\ -1 & 4 & -1 \\ 0 & -1 & 0 \end{bmatrix}, \qquad (13)$$

with no dependency on $h$. The continuous Laplacian is typically discretized with a grid constant $1/h^2$, to yield a discrete operator $\Delta_h$, i.e., $\Delta \approx \Delta_h = -\frac{1}{h^2}\hat{\mathbf{L}}$. Using a coarser grid with a step size $2h$, for example, yields $\Delta \approx \Delta_{2h} = -\frac{1}{4h^2}\hat{\mathbf{L}}$. That means that if we coarsen the grid by a factor of two, the effective discrete Laplacian operator coefficients decrease by a factor of four. Hence, the value of $\kappa$ in Equation (12) multiplies by four, resulting in a spatial distribution of Equation (12) that spreads faster for the coarser representation as also illustrated in Figure 4.

**Interleaving Scales.** The discussion above showed how utilizing coarser scales helps in expanding the ERF. However, in many tasks, ERF is not the only property that is needed for a GNN to obtain favorable performance. By using coarser scales (via pooling), we also lose information in our feature maps–mostly information that is given in high-frequency feature maps. Hence, to obtain the best performance we must combine both high and low-resolution maps and graph operation to capture and fuse high- and low-frequency information, and have a high ERF. We obtain this by applying network layers that use multiple graph levels in each block and fuse the information between them.

### 4.2. Interleaved Multiscale Message-Passing Networks

We now describe our IM-MPNN, which involves three main parts: (i) an MPNN backbone; (ii) pooling and unpooling layers; and (iii) an interleaving mechanism.

**MPNN Backbone.** Our IM-MPNN offers a generalist approach for enhancing the ERF of existing and future MPNNs by utilizing interleaved multiscale representations of input graphs. Therefore, we consider the following general update rule for node features $\mathbf{x}^{(\ell)}$:

$$\mathbf{x}^{(\ell+1)} = \text{MPNN}^{(\ell)}(\mathbf{x}^{(\ell)}; \mathbf{A}), \qquad (14)$$

where $\text{MPNN}^{(\ell)}$ denotes the MPNN backbone used at the $\ell$-th layer. In our experiments in Section 5, we demonstrate the effectiveness of our IM-MPNN on several backbones.

**Pooling and Unpooling.** The second component of IM-MPNN involves the pooling and unpooling operations, which are essential to obtain a multiscale representation of input graphs and their features. Given a graph $\mathcal{G} = (\mathcal{V}, \mathcal{E})$, we define the pooling operation to be an aggregation of node pairs in a pairing set $P$. That is, the node pairing $P$ represents node pairs $\{v_i, v_j\}$, that are aggregated into a

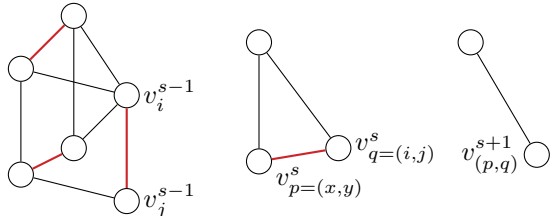

Figure 5: Coarsening of a graph according to a given pairing (edges marked in red).

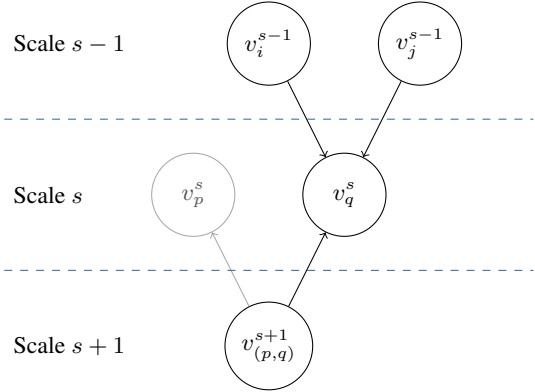

Figure 6: Scale-mix. Each node receives information from its parents' and children's nodes of consecutive scales.

new node $v'_q$ in the pooled graph. More formally, the pooled graph $\mathcal{G}' = (\mathcal{V}', \mathcal{E}')$ is given by

$$
\begin{aligned}
\mathcal{V}' &= \{v'_q \mid v'_q = \{v_i, v_j\} \in P\}, \\
\mathcal{E}' &= \{(v'_q, v'_p) \mid \exists v_i \in v'_q, v_j \in v'_q, (v_i, v_j) \in \mathcal{E}\},
\end{aligned}
\tag{15}
$$

i.e., the edges in the pooled graph represent edges that existed in the original graph between components of node pairs in the pooled graph. Note that the procedure described in Equation (15) denotes a *single* pooling step, between two consecutive scales. Overall, we denote the transition between the $s$-th and the $s+1$-th scales as follows:

$$
\mathbf{x}^{s+1}, \mathbf{A}^{s+1} = \text{POOL}(\mathbf{x}^s, \mathbf{A}^s; P^s),
\tag{16}
$$

where $\mathbf{x}^{s+1} \in \mathbb{R}^{n_{s+1} \times c}, \mathbf{A}^{s+1} \in \mathbb{R}^{n_{s+1} \times n_{s+1}}$, and $n_{s+1} = |P^s| < n_s$. That is, the POOL function takes the node features and adjacency matrix at the $s$-th level, and decreases the number of nodes to $n_s$ according to $P$ and some aggregation function (e.g., mean), and returns a coarsened representation of the node features and the adjacency matrix. In practice, we use the Graclus algorithm (Dhillon et al., 2007) to find pairs of nodes to be averaged, as follows:

$$
\mathbf{x}^{s+1}_{\{v_i, v_j\}} = \frac{1}{2}(\mathbf{x}^s_i + \mathbf{x}^s_j).
\tag{17}
$$

Analogously, the *unpooling* layer applies the opposite operation. Using the same node pairing $P^s$ of the corresponding

pooling operation, we revert the graph to the previous size:

$$
\hat{\mathbf{x}}^s, \hat{\mathbf{A}}^s = \text{UNPOOL}(\mathbf{x}^{s+1}, \mathbf{A}^{s+1}; P^s),
\tag{18}
$$

where $\hat{\mathbf{x}}^s \in \mathbb{R}^{n_s \times c}, \hat{\mathbf{A}}^s \in \mathbb{R}^{n_s \times n_s}$. In practice, we distribute the feature at a pooled node in $\mathbf{x}^{s+1}$ to its two corresponding nodes in the finer representation in $\hat{\mathbf{x}}^s$.

We note that it is possible to stack several pooling and unpooling layers, as we do in our IM-MPNN, illustrated in Figure 2. To be specific, we coarsen the graph $S$ times. The repeated pooling yields a set of graphs $\{\mathcal{G}_s\}_{s=0}^S$, where $\mathcal{G}_0 = \mathcal{G}$ is the original input graph, and $\mathcal{G}^S$ is the coarsest graph. We denote $v^s$ as a node from $\mathcal{G}_s$, and $\mathbf{A}^s$ as the adjacency matrix of $\mathcal{G}_s$. Note that each node $v^s \in \mathcal{V}_s, s > 0$ is a coarsening of several nodes $v_1^{s-1}, v_2^{s-1}, \dots, v_k^{s-1} \in \mathcal{V}_{s-1}$, creating a relationship between nodes at consecutive scales, as shown in Figure 5.

**Multiscale Interleaving.** As illustrated in Figure 2, the $S$ scales of the input graph $\mathcal{G}$ are maintained throughout the network, each with its own message-passing layer and a separate set of weights. That is, Equation (14) for scale $s$ reads the following update rule:

$$
\tilde{\mathbf{x}}^{(s,\ell)} = \text{MPNN}^{(s,\ell)}(\mathbf{x}^{(s,\ell)}, \mathbf{A}^s).
\tag{19}
$$

However, without further modifications, those graphs will be processed separately. Therefore, we define a multiscale interleaving mechanism, where relevant nodes of consecutive scales share information. Specifically, let us consider a node $v^s_{q=(i,j)}$ from the $s$-th scale, and its related node $v^{s+1}_{(p,q)}$ from the $(s+1)$-th scale, and $v_i^{s-1}, v_j^{s-1}$ from the $(s-1)$-th scale, as illustrated in Figure 6. We define:

$$
\mathbf{x}^{(s,\ell+1)}_{q=(i,j)} =
$$
$$
\tilde{\mathbf{x}}^{(s,\ell)}_{q=(i,j)} + W^{(s,\ell)}_{l2h} \frac{1}{2}(\tilde{\mathbf{x}}^{(s-1,\ell)}_i + \tilde{\mathbf{x}}^{(s-1,\ell)}_j) + W^{(s,\ell)}_{h2l} \tilde{\mathbf{x}}^{(s+1,\ell)}_{(p,q)},
\tag{20}
$$

where $W^{(s,\ell)}_{l2h}, W^{(s,\ell)}_{h2l}$ are learnable weights for the lower-to-higher and higher-to-lower information passing.

Lastly, before the final head (classifier), we cast all the information back to the original graph nodes, by concatenating the features of the coarse nodes into their source node, which can be done by recursive unpooling and concatenating

$$
\mathbf{z}^{(L,s)} = [\mathbf{x}^{(L,s)}, \text{UNPOOL}(\mathbf{z}^{(L,s+1)}, \mathbf{A}^{s+1}; P^s)],
\tag{21}
$$

for $s = S - 1, \dots, 0$, where we start from $\mathbf{z}^{(L,S)} = \mathbf{x}^{(L,S)}$.

### 4.3. Complexity of IM-MPNN

As described, a single IM-MPNN layer is constructed out of two steps, the first being an MPNN operation on each scale, and then a scale-mix operation. An MPNN operation

Table 1: Multiscale version of different GCNs for PascalVOC-SP and COCO-SP, increasing scales while reducing the width to stay within 500k params budget.

| Method | scales | PascalVOC-SP Test F1 ↑ | COCO-SP Test F1 ↑ |
|---|---|---|---|
| GCN | – | $0.2078_{\pm0.0031}$ | $0.1338_{\pm0.0007}$ |
| IM-GCN | 1 | $0.2672_{\pm0.0045}$ | $0.1655_{\pm0.0013}$ |
| IM-GCN | 2 | $0.2846_{\pm0.0084}$ | $0.1787_{\pm0.0016}$ |
| IM-GCN | 3 | $0.2910_{\pm0.0058}$ | $0.1896_{\pm0.0009}$ |
| IM-GCN | 4 | $\mathbf{0.2929}_{\pm0.0058}$ | $\mathbf{0.1960}_{\pm0.0023}$ |
| GINE | – | $0.2718_{\pm0.0054}$ | $0.2125_{\pm0.0009}$ |
| IM-GINE | 1 | $0.2757_{\pm0.0013}$ | $0.2289_{\pm0.0009}$ |
| IM-GINE | 2 | $0.2873_{\pm0.0055}$ | $0.2387_{\pm0.0016}$ |
| IM-GINE | 3 | $0.2909_{\pm0.0074}$ | $\mathbf{0.2475}_{\pm0.0017}$ |
| IM-GINE | 4 | $\mathbf{0.2975}_{\pm0.0088}$ | $0.2472_{\pm0.0037}$ |
| GatedGCN | – | $0.3880_{\pm0.0040}$ | $0.2922_{\pm0.0018}$ |
| IM-GatedGCN | 1 | $0.4180_{\pm0.0062}$ | $0.3294_{\pm0.0021}$ |
| IM-GatedGCN | 2 | $0.4297_{\pm0.0043}$ | $0.3453_{\pm0.0017}$ |
| IM-GatedGCN | 3 | $\mathbf{0.4332}_{\pm0.0045}$ | $\mathbf{0.3501}_{\pm0.0033}$ |
| IM-GatedGCN | 4 | $0.4317_{\pm0.0078}$ | $0.3414_{\pm0.0061}$ |

Table 2: Multiscale version of different GCNs for Peptides-func and Peptides-struct, increasing scales while reducing the width to stay within 500k params budget.

| Method | scales | Peptides-func Test AP ↑ | Peptides-struct Test MAE ↓ |
|---|---|---|---|
| GCN | – | $0.6860_{\pm0.0050}$ | $0.2460_{\pm0.0007}$ |
| IM-GCN | 1 | $0.6822_{\pm0.0088}$ | $\mathbf{0.2453}_{\pm0.0009}$ |
| IM-GCN | 2 | $0.6936_{\pm0.0074}$ | $0.2473_{\pm0.0018}$ |
| IM-GCN | 3 | $\mathbf{0.6942}_{\pm0.0083}$ | $0.2498_{\pm0.0025}$ |
| IM-GCN | 4 | $0.6907_{\pm0.0053}$ | $0.2473_{\pm0.0006}$ |
| GINE | – | $0.6621_{\pm0.0067}$ | $0.2473_{\pm0.0017}$ |
| IM-GINE | 1 | $0.6745_{\pm0.0022}$ | $0.2477_{\pm0.0007}$ |
| IM-GINE | 2 | $0.6884_{\pm0.0027}$ | $\mathbf{0.2464}_{\pm0.0008}$ |
| IM-GINE | 3 | $0.6948_{\pm0.0034}$ | $0.2489_{\pm0.0009}$ |
| IM-GINE | 4 | $\mathbf{0.6959}_{\pm0.0021}$ | $0.2477_{\pm0.0015}$ |
| GatedGCN | – | $0.6765_{\pm0.0047}$ | $0.2477_{\pm0.0009}$ |
| IM-GatedGCN | 1 | $0.6713_{\pm0.0051}$ | $0.2456_{\pm0.0010}$ |
| IM-GatedGCN | 2 | $0.6714_{\pm0.0046}$ | $0.2469_{\pm0.0009}$ |
| IM-GatedGCN | 3 | $\mathbf{0.6830}_{\pm0.0037}$ | $\mathbf{0.2454}_{\pm0.0014}$ |
| IM-GatedGCN | 4 | $0.6773_{\pm0.0041}$ | $0.2455_{\pm0.0006}$ |

complexity is $\mathcal{O}(|V|+|E|)$, however, each of the operations is performed over a downscaled graph with a factor of 2. Hence, the time complexity of performing MPNN on all the scales (including the 0-th scale) is

$$\sum_{s=0}^{S} \mathcal{O}\left(\frac{|V|}{2^s} + \frac{|E|}{2^s}\right) = \mathcal{O}\left(2\left(|V| + |E|\right)\right). \quad (22)$$

The scale-mix operation incorporates a calculation that involves 4 nodes for every node on every scale, hence the time complexity is given by

$$\sum_{s=0}^{S} \mathcal{O}\left(\frac{|V|}{2^s}\right) = \mathcal{O}\left(2|V|\right). \quad (23)$$

Hence, the time complexity of IM-MPNN is $\mathcal{O}\left(|V| + |E|\right)$.

# 5. Experimental Results

**Objectives.** We evaluate IM-MPNN and compare it with various methods. Specifically, we seek to address the following questions: (i) How effective is IM-MPNN at propagating information to distant nodes? And how well it predicts graph properties related to long-range interactions? (Section 5.2) (ii) How does IM-MPNN perform on real-world long-range benchmarks? (Sections 5.1, 5.3 and 5.4) (iii) How well does IM-MPNN perform with different message-passing protocols? (Sections 5.1 and 5.4). Additional experiments are available in Appendix E.

**Baselines.** We evaluate IM-MPNN with several message-passing protocols with linear complexity (similar to the

complexity of IM-MPNN) and measure the improvement over them: (i) GCN (Kipf & Welling, 2016), (ii) GatedGCN (Bresson & Laurent, 2017), (iii) GINE (Hu et al., 2020), (iv) and GAT (Veličković et al., 2018).

## 5.1. Long Range Graph Benchmark

**Setup.** We test our method on Long Range Graph Benchmark (Dwivedi et al., 2022) following the improved evaluation (Tönshoff et al., 2023). The benchmark is constructed of several datasets with long-range dependencies and, therefore, can be difficult for MPNNs. We take the same reported MPNN architecture, namely GCN (Kipf & Welling, 2016), GINE (Hu et al., 2020), and GatedGCN (Bresson & Laurent, 2017), and modify the architecture using our multiscale approach with different scales. Since an additional scale introduces more parameters, we reduce the channel dimension in order to stay within a budget of 500k parameters.

**Results.** We see a significant increase in scores, of up to 41% relative improvement on both Pascalvoc-SP and COCO-SP, see Table 1. While peptide-func and peptide-struct (Table 2) are less affected by different configurations, we still see a slight advantage in using our multiscale architecture.

## 5.2. Graph Transfer

**Setup.** We consider a graph transfer task (Di Giovanni et al., 2023), where the goal is to transfer a label from a source to a target node with a distance of $k$ hops, using a network of depth $k$. We note that this task can be effectively solved only by non-dissipative methods that preserve source information.

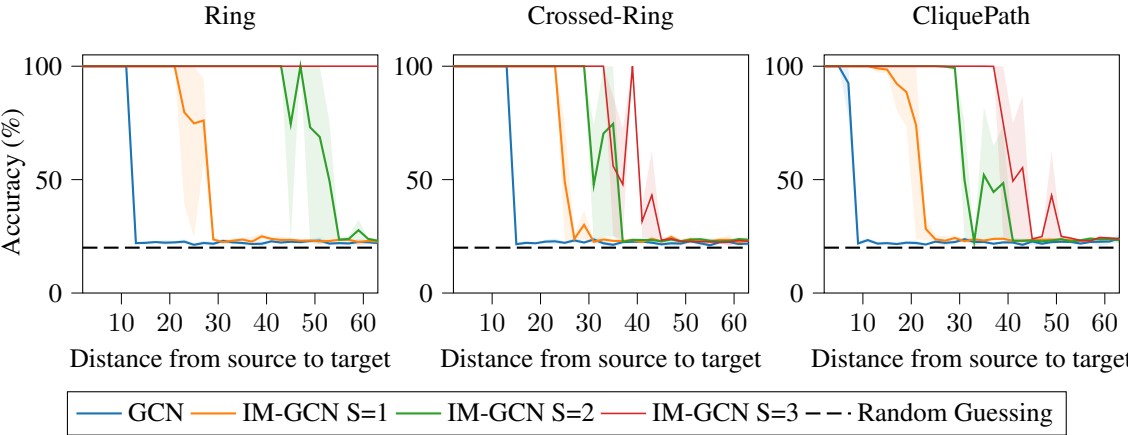

Figure 7: Graph transfer results over three graph types (ring, crossed-ring, and cliquepath). The network depth (i.e. number of layers) is the same as the distance between the source and the target node. We see that adding more scales to IM-GCN improves the ability of the network to transfer information across long distances.

We consider three graph distributions, i.e., ring, crossed-ring, and clique-path, with distances $k = \{2, \dots 65\}$. Increasing $k$ increases the task complexity. When using IM-GCN, we decrease the number of channels according to the number of scales for a fair comparison. Appendix C.1 provides additional details about the task and the datasets.

**Results.** Figure 7 reports the results on the graph transfer tasks. Overall, compared to GCN, IM-GCN show an increase in the ability to transfer information over long distances. E.g., for CliquePath, where GCN start failing at around $k = 7$ while IM-GCN with scales=3 still have 100% accuracy at around $k = 37$.

### 5.3. City-Networks

**Setup.** We evaluate IM-MPNN on the City-Networks benchmark (Liang et al., 2025), a recently introduced large-scale dataset based on real-world city road networks designed to test long-range dependencies in graph learning (see Appendix C.3 for details). Derived from real-world city road maps (Paris, Shanghai, Los Angeles, and London) using OpenStreetMap data, the graphs feature between 100k and 570k nodes with significantly larger diameters (over 100–400) compared to traditional datasets. Following the original setup, we train each model for 20k epochs using AdamW with a learning rate of $10^{-3}$ and a weight decay of $10^{-5}$, and do not perform any additional hyperparameter tuning across runs to maintain consistency with the baseline protocol.

**Results.** The results are reported in Table 3. When using IM-GCN, we see a significant increase in accuracy compared to the GCN baseline of between +10% and +16.5%, when using the IM-MPNN version of the network.

Table 3: Results of test accuracy ($\uparrow$) on City-Networks.

| Method | scales | Paris | Shanghai | Los Angeles | London |
|---|---|---|---|---|---|
| GCN | – | $47.3_{\pm 0.2}$ | $52.4_{\pm 0.3}$ | $45.9_{\pm 1.0}$ | $43.8_{\pm 0.3}$ |
| IM-GCN | 1 | $54.4_{\pm 0.2}$ | $63.4_{\pm 0.8}$ | $59.8_{\pm 0.2}$ | $53.8_{\pm 0.3}$ |
| IM-GCN | 2 | $54.9_{\pm 0.2}$ | $65.8_{\pm 0.4}$ | $61.6_{\pm 0.2}$ | $57.1_{\pm 0.1}$ |
| IM-GCN | 3 | $\mathbf{55.3}_{\pm 0.3}$ | $66.7_{\pm 0.3}$ | $62.2_{\pm 0.2}$ | $58.4_{\pm 0.1}$ |
| IM-GCN | 4 | $54.9_{\pm 0.2}$ | $\mathbf{67.8}_{\pm 0.1}$ | $\mathbf{62.4}_{\pm 0.3}$ | $\mathbf{58.9}_{\pm 0.1}$ |

### 5.4. Heterophilic Node Classification

**Setup.** We evaluate IM-MPNN on five heterophilic node classification benchmarks introduced by Platonov et al. (2023): Roman-Empire, Amazon-Ratings, Minesweeper, Tolokers, and Questions. The datasets span a variety of domains, from natural language and e-commerce to crowd-sourcing and synthetic grids, each exhibiting weak homophily and requiring models to reason beyond local neighborhoods. We follow the original experimental setup, training models with the AdamW optimizer for up to 300 epochs on the official splits, without additional hyperparameter tuning. Additional details are provided in Appendix C.4.

**Results.** We benchmark IM-MPNN with results from Platonov et al. (2023); Finkelshtein et al. (2024); Behrouz & Hashemi (2024); Müller et al. (2024) and find it achieves competitive performance, often surpassing state-of-the-art methods. Our results are reported in Table 4, with additional comparisons in Table 6. Notably, the competitive performance on larger graphs and complex heterophilic scenarios while retaining the linear complexity of standard MPNNs, further highlights its effectiveness compared to Graph Transformers and heterophily-designated GNNs.

Table 4: Mean test set score and standard deviation are averaged over four random initializations on heterophilic datasets. **First**, **second**, and **third** best results per task are color-coded. Additional comparisons are provided in Table 6.

| Model | Roman-empire Accuracy ↑ | Amazon-ratings Accuracy ↑ | Minesweeper AUC ↑ | Tolokers AUC ↑ | Questions AUC ↑ |
|---|---|---|---|---|---|
| **Graph Transformers** | | | | | |
| Exphormer | $89.03_{\pm 0.37}$ | $53.51_{\pm 0.46}$ | $90.74_{\pm 0.53}$ | $83.77_{\pm 0.78}$ | $73.94_{\pm 1.06}$ |
| GOAT | $71.59_{\pm 1.25}$ | $44.61_{\pm 0.50}$ | $81.09_{\pm 1.02}$ | $83.11_{\pm 1.04}$ | $75.76_{\pm 1.66}$ |
| GPS$_{\text{GCN+Performer}}$ (RWSE) | $84.72_{\pm 0.65}$ | $48.08_{\pm 0.85}$ | $92.88_{\pm 0.50}$ | $84.81_{\pm 0.86}$ | $76.45_{\pm 1.51}$ |
| GPS$_{\text{GAT+Performer}}$ (RWSE) | $87.04_{\pm 0.58}$ | $49.92_{\pm 0.68}$ | $91.08_{\pm 0.58}$ | $84.38_{\pm 0.91}$ | $77.14_{\pm 1.49}$ |
| GT-sep | $87.32_{\pm 0.39}$ | $52.18_{\pm 0.80}$ | $92.29_{\pm 0.47}$ | $82.52_{\pm 0.92}$ | $78.05_{\pm 0.93}$ |
| **Heterophily-Designated GNNs** | | | | | |
| FSGNN | $79.92_{\pm 0.56}$ | $52.74_{\pm 0.83}$ | $90.08_{\pm 0.70}$ | $82.76_{\pm 0.61}$ | $78.86_{\pm 0.92}$ |
| GBK-GNN | $74.57_{\pm 0.47}$ | $45.98_{\pm 0.71}$ | $90.85_{\pm 0.58}$ | $81.01_{\pm 0.67}$ | $74.47_{\pm 0.86}$ |
| JacobiConv | $71.14_{\pm 0.42}$ | $43.55_{\pm 0.48}$ | $89.66_{\pm 0.40}$ | $68.66_{\pm 0.65}$ | $73.88_{\pm 1.16}$ |
| **MPNNs** | | | | | |
| GCN | $73.69_{\pm 0.74}$ | $48.70_{\pm 0.63}$ | $89.75_{\pm 0.52}$ | $83.64_{\pm 0.67}$ | $76.09_{\pm 1.27}$ |
| Gated-GCN | $74.46_{\pm 0.54}$ | $43.00_{\pm 0.32}$ | $87.54_{\pm 1.22}$ | $77.31_{\pm 1.14}$ | $76.61_{\pm 1.13}$ |
| GAT | $80.87_{\pm 0.30}$ | $49.09_{\pm 0.63}$ | $92.01_{\pm 0.68}$ | $83.70_{\pm 0.47}$ | $77.43_{\pm 1.20}$ |
| GAT-sep | $88.75_{\pm 0.41}$ | $52.70_{\pm 0.62}$ | $93.91_{\pm 0.35}$ | $83.78_{\pm 0.43}$ | $76.79_{\pm 0.71}$ |
| CO-GNN$(\Sigma, \Sigma)$ | $91.57_{\pm 0.32}$ | $51.28_{\pm 0.56}$ | $95.09_{\pm 1.18}$ | $83.36_{\pm 0.89}$ | $80.02_{\pm 0.86}$ |
| CO-GNN$(\mu, \mu)$ | $91.37_{\pm 0.35}$ | $54.17_{\pm 0.37}$ | $97.31_{\pm 0.41}$ | $84.45_{\pm 1.17}$ | $76.54_{\pm 0.95}$ |
| **Interleaved Multiscale (Ours)** | | | | | |
| IM-GCN | $83.53_{\pm 0.57}$ | $52.37_{\pm 0.66}$ | $91.80_{\pm 0.58}$ | $84.17_{\pm 0.71}$ | $78.17_{\pm 0.89}$ |
| IM-GatedGCN | $90.82_{\pm 0.59}$ | $54.01_{\pm 0.27}$ | $97.32_{\pm 0.83}$ | $85.10_{\pm 0.84}$ | $79.27_{\pm 0.91}$ |
| IM-GAT | $84.48_{\pm 0.32}$ | $51.16_{\pm 0.55}$ | $92.68_{\pm 0.49}$ | $85.21_{\pm 0.43}$ | $77.98_{\pm 1.01}$ |
| IM-GAT-sep | $89.93_{\pm 0.34}$ | $53.97_{\pm 0.58}$ | $96.15_{\pm 0.37}$ | $85.44_{\pm 0.40}$ | $77.92_{\pm 0.83}$ |
| IM-CO-GNN$(\Sigma, \Sigma)$ | $92.08_{\pm 0.33}$ | $53.11_{\pm 0.59}$ | $95.79_{\pm 0.96}$ | $85.25_{\pm 1.03}$ | $80.49_{\pm 0.92}$ |
| IM-CO-GNN$(\mu, \mu)$ | $92.00_{\pm 0.41}$ | $54.43_{\pm 0.41}$ | $97.39_{\pm 0.35}$ | $85.77_{\pm 1.05}$ | $78.92_{\pm 0.87}$ |

## 6. Conclusion

In this work, we have presented a novel approach, Interleaved Multiscale Message-Passing Neural Networks (IM-MPNN), that effectively addresses the limitations of MPNNs in capturing long-range dependencies and their limited effective receptive field (ERF). By enhancing the ERF of MPNNs while maintaining their linear computational complexity in graph size, our method introduces hierarchical coarsening and scale-mixing mechanisms to extend the receptive field of GNNs without sacrificing efficiency.

Through extensive theoretical and experimental evaluations, we demonstrated the advantages of our method across diverse datasets, including benchmarks with long-range dependencies, heterophilic settings, and large-scale graphs. Our results consistently highlight the superior performance of IM-MPNN in both predictive accuracy and computational efficiency, even when compared to state-of-the-art methods like Graph Transformers and heterophily-designated GNNs.

## Acknowledgments

This work was supported in part by Grant No 2023771 from the United States-Israel Binational Science Foundation (BSF), by Grant No 2411264 from the United States National Science Foundation (NSF), by Israel Science Foundation Personal Grant #360/21, by the Lynn and William Frankel Center at BGU CS, and by the Israeli Council for Higher Education (CHE) via the Data Science Research Center at BGU. S.E.F.'s work was supported by the BGU's Hi-Tech Scholarship. S.E.F.'s and R.S.W.'s work was also supported by the Kreitman School of Advanced Graduate Studies. M.E. is funded by the Blavatnik-Cambridge fellowship, the Cambridge Accelerate Programme for Scientific Discovery, and the Maths4DL EPSRC Programme.

## Impact Statement

This paper presents a method for improving the limited ERF in GNNs. Generally, this improves the performance of existing standard GNNs and may contribute to any field in which unstructured data is processed using GNNs. We are not aware of possible negative societal impacts of our research.

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

## A. On Graclus

The Graclus algorithm is an efficient, non-parametric graph clustering method commonly utilized for hierarchical pooling in graph neural networks (GNNs). Originally, Graclus employs a multilevel clustering strategy—consisting of graph coarsening by iteratively merging node pairs based on edge weights, performing initial clustering on the reduced graph, and refining the partitions during successive uncoarsening steps. This process efficiently approximates graph partitioning objectives like normalized cut without computationally intensive eigenvector computations.

In practice, particularly within frameworks such as PyTorch Geometric (PyG), a simplified version of Graclus is used. PyG's implementation performs a single-pass greedy clustering, pairing each node with its highest-weight neighbor to approximately halve the graph size at each pooling step. This variant omits the refinement phase and the broader multilevel approach of the original algorithm, prioritizing computational efficiency and GPU compatibility. In IM-MPNN specifically, we further simplify by using this PyG implementation in an unweighted manner, clustering node pairs without considering edge weights.

It is important to note that IM-MPNN does not rely on the coarsening algorithm and can be easily modified to work with any alternative for the simplified Graclus clustering.

## B. Choosing the Number of Scales

The number of scales used in IM-MPNN is a hyperparameter that requires empirical tuning, analogous to other architectural parameters such as depth and width. However, intuitive guidance can be drawn from structural and attribute properties of the input graph. Graphs with larger diameters naturally benefit from additional coarsening levels, as these allow the model to capture interactions occurring across longer distances by aggregating information into nodes at progressively coarser scales. Conversely, once nodes at the coarsest scale represent substantial portions of the graph, further increases in the number of scales typically yield diminishing returns. Another relevant factor is the attribute homophily of the graph: graphs with high homophily, where nodes predominantly connect to similar nodes, might require fewer scales because local neighborhoods already provide substantial predictive information. In contrast, heterophilic graphs—where informative node interactions are more dispersed—could benefit from additional scales, enabling effective aggregation of meaningful signals from more distant regions.

## C. Datasets and Experimental Settings

### C.1. Graph Transfer

**Dataset.** The graph transfer task follows the settings of Di Giovanni et al. (2023). In each graph, a one-hot label is assigned to a target node at a distance $k$ from the source, and a constant unitary feature vector is assigned to all other nodes. Graphs were sampled from three different distributions: ring, crossed-ring, and clique-path (see Figure 8 for a visual exemplification). In ring graphs, the nodes form a cycle of size $n$, with the source and target placed $\lfloor n/2 \rfloor$ apart. Similarly, crossed-ring graphs consisting of cycles of size $n$, but introduced additional edges crossing intermediate nodes, while still maintaining a source-target distance of $\lfloor n/2 \rfloor$. Lastly, the clique-path graph contains a clique of size $\lfloor n/2 \rfloor$ followed by a path of length $\lfloor n/2 \rfloor$. The source node is placed in the clique and the target is at the other end of the path. Our experiments focus on a regression task aimed at assigning the target label of the target to the source node. In all graphs, we refer to the distance between the source node and the target node as $k$ regardless of the number of nodes in the graph.

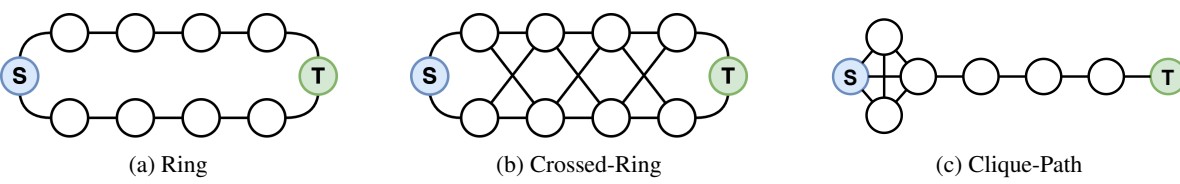

(a) Ring        (b) Crossed-Ring        (c) Clique-Path

Figure 8: Line, ring, and crossed-ring graphs where the distance between source and target nodes is equal to 5. Nodes marked with "S" are source nodes, while the nodes with a "T" are target nodes.

**Experimental Setting.**    We design each model as a combination of three main components. The first is the encoder which maps the node input features into a latent hidden space; the second is the graph convolution (i.e., IM-MPNN or the other baselines); and the third is a readout that maps the output of the convolution into the output space. The encoder and the readout share the same architecture among all models in the experiments.

We use the same hyperparameters given by Di Giovanni et al. (2023), with the only difference being reducing the number of channels when increasing the number of scales for a fair comparison (i.e., increased width might help against the other causes of over-squashing). For each model configuration, we perform 3 training runs with different weight initialization and report the average and standard deviation of the results.

### C.2. Long Range Graph Benchmark

**Dataset.**    To assess the performance on real-world long-range graph benchmarks, we considered the Pascalvoc-SP, COCO-SP, Peptides-func, and Peptides-struct datasets from the Long Range Graph Benchmark (Dwivedi et al., 2022).

The Pascalvoc-SP and COCO-SP datasets are based on Pascal VOC 2011 image dataset (Everingham et al., 2010) and MS COCO image dataset (Lin et al., 2014), where each image is divided into coherent regions called superpixels based on the SLIC algorithm (Achanta et al., 2010). Each image is then a graph whose nodes are its superpixels which are connected with an edge if they share a boundary. The task of both datasets is node classification which predicts the label of the semantic segmentation of the original datasets. Pascalvoc-SP contains 11,355 graphs, with a total of 5.4 million nodes. COCO-SP contains 123,286 graphs, with a total of 58.8 million nodes.

The peptides-related graphs represent 1D amino acid chains, with nodes corresponding to the heavy (non-hydrogen) atoms of the peptides, and edges representing the bonds between them. Peptides-func is a multi-label graph classification dataset containing 10 classes based on peptide functions, such as antibacterial, antiviral, and cell-cell communication. Peptides-struct is a multi-label graph regression dataset, focused on predicting 3D structural properties of peptides. The regression tasks involve predicting the inertia of molecules based on atomic mass and valence, the maximum atom-pair distance, sphericity, and the average distance of all heavy atoms from the plane of best fit. Both datasets, Peptides-func and Peptides-struct, consist of 15,535 graphs, encompassing a total of 2.3 million nodes.

For all datasets, we used the official splits as by Dwivedi et al. (2022), and reported the average and standard-deviation performance across 3 seeds.

**Experimental Setting.**    We employ the same datasets and experimental setting presented by Tönshoff et al. (2023), an improved hyperparameter tuning for the LRGB dataset. We used the reported hyperparameters, changing the number of scales while using our IM-MPNN variant, and only modifying the number of channels to accommodate for the change in the number of parameters in order to stay within the 500K parameter budget. For each model configuration, we perform 3 training runs with different weight initialization and report the average of the results.

### C.3. City-Networks Benchmark

**Dataset.** City-Networks (Liang et al., 2025) is a recently introduced benchmark specifically designed to evaluate the ability of graph neural networks (GNNs) to capture long-range dependencies. Traditional graph learning benchmarks often involve graphs with small diameters and relatively limited structural complexity, where most tasks can be solved using information aggregated from immediate or short-range neighbors. In contrast, City-Networks provides a more challenging setting that demands explicit modeling of long-range interactions across large graphs.

The dataset is constructed from real-world road networks of four major cities—Paris, Shanghai, Los Angeles, and London—using data extracted from OpenStreetMap. Each graph contains between 100,000 and 570,000 nodes, making them several orders of magnitude larger than common benchmark graphs. Furthermore, these graphs exhibit very large diameters, typically ranging from 100 to over 400, ensuring that many node pairs are separated by long paths. This structural property makes City-Networks particularly suitable for assessing how well a GNN can propagate information across extended distances.

Each node in the dataset is associated with a set of geographic and road-related features, such as spatial coordinates, road type, and additional local attributes derived from the map data. The labels are not based on external metadata but are generated using an approximation of node eccentricity—the maximum distance from a node to any other node in the graph. This task setup emphasizes the need for models to gather and process information from distant nodes, rather than relying

solely on local neighborhood structures.

Another notable aspect of City-Networks is its transductive nature: models are trained and evaluated on the same graph, as opposed to inductive settings where generalization to unseen graphs is required. This design choice aligns with the benchmark's primary goal of probing long-range information flow within a fixed large structure, rather than testing generalization across different domains.

**Experimental Setting.** We follow Liang et al. (2025) training procedure of 20k epochs, batch size of 20k, learning rate of $10^{-3}$, and weight decay of $10^{-5}$. The model is evaluated for accuracy every 100 epochs, and the model with the best validation is saved for final testing. All the scenarios are repeated 5 times, and we report their means and standard deviations.

### C.4. Heterophilic Node Classification

**Dataset.** We evaluate IM-MPNN on the Roman-Empire, Amazon-Ratings, Minesweeper, Tolokers, and Questions tasks from (Platonov et al., 2023). *Roman-Empire* is derived from Wikipedia, where nodes represent words and edges connect syntactically related or sequential words. The task involves classifying words into 18 syntactic roles. The graph is chain-like with sparse connectivity and long-range dependencies. *Amazon-Ratings* models an Amazon product co-purchasing network. Nodes represent products, edges connect frequently co-purchased items, and the goal is to predict average product ratings (five classes). Node features are fastText embeddings of product descriptions. *Minesweeper* is a synthetic dataset based on a 100x100 grid. Nodes represent cells, edges connect neighbors, and 20% of nodes are designated as mines. The task is to predict mine locations using one-hot-encoded neighboring mine counts as features. *Tolokers* is based on the Toloka crowdsourcing platform (Likhobaba et al., 2023). Nodes represent workers (tolokers), edges indicate collaboration, and the task is to predict whether a worker has been banned, using profile and performance features. *Questions* originates from the Yandex Q question-answering platform. Nodes represent users, edges connect those who have interacted by answering each other's questions, and the goal is to predict user retention. Node features are derived from user descriptions. A summary of the dataset statistics is provided in Table 5.

**Experimental Setting.** We follow the training and evaluation protocols from Platonov et al. (2023), and in particular follow the same splits. For hyperparameters, we consider learning rates and weight decays in the range of $1e-5$ to $1e-3$ using the AdamW optimizer, and we consider $2, 3, 4, 8$ scales within our IM-MPNN framework.

**Additional Comparisons.** In addition to the benchmarking provided in Section 5, and in particular in Table 4, we provided additional comparisons with more baselines in Table 6, showing that also in this case, our IM-MPNN offers similar or better performance, while maintaining linear complexity with respect to the graph number of nodes and edges. We consider a range of models, including classical MPNN-based methods such as GCN (Kipf & Welling, 2016), GraphSAGE (Hamilton et al., 2017), GAT (Veličković et al., 2018), GatedGCN (Bresson & Laurent, 2017), GIN (Xu et al., 2019), GINE (Hu et al., 2020), and CoGNN (Finkelshtein et al., 2024); heterophily-specific models like H2GCN (Zhu et al., 2020), CPGNN (Zhu et al., 2021), FAGCN (Bo et al., 2021), GPR-GNN (Chien et al., 2021), FSGNN (Maurya et al., 2022), GloGNN (Li et al., 2022), GBK-GNN (Du et al., 2022), and JacobiConv (Wang & Zhang, 2022); Graph Transformers such as Transformer (Dwivedi & Bresson, 2021), GT (Shi et al., 2021), SAN (Kreuzer et al., 2021), GPS (Rampášek et al., 2022), GOAT (Kong et al., 2023), and Exphormer (Shirzad et al., 2023); Higher-Order DGNs like DIGL (Gasteiger et al., 2019), MixHop (Abu-El-Haija et al., 2019), and DRew (Gutteridge et al., 2023).

Table 5: Statistics of the heterophilic node classification datasets.

|  | Roman-empire | Amazon-ratings | Minesweeper | Tolokers | Questions |
|---|---|---|---|---|---|
| N. nodes | 22,662 | 24,492 | 10,000 | 11,758 | 48,921 |
| N. edges | 32,927 | 93,050 | 39,402 | 519,000 | 153,540 |
| Avg degree | 2.91 | 7.60 | 7.88 | 88.28 | 6.28 |
| Diameter | 6,824 | 46 | 99 | 11 | 16 |
| Node features | 300 | 300 | 7 | 10 | 301 |
| Classes | 18 | 5 | 2 | 2 | 2 |
| Edge homophily | 0.05 | 0.38 | 0.68 | 0.59 | 0.84 |

Table 6: Mean test set score and standard deviation are averaged over four random weight initializations on heterophilic datasets (higher is better).

| Model | Roman-empire Accuracy ↑ | Amazon-ratings Accuracy ↑ | Minesweeper AUC ↑ | Tolokers AUC ↑ | Questions AUC ↑ |
|---|---|---|---|---|---|
| **MPNNs** | | | | | |
| GAT | $80.87_{\pm 0.30}$ | $49.09_{\pm 0.63}$ | $92.01_{\pm 0.68}$ | $83.70_{\pm 0.47}$ | $77.43_{\pm 1.20}$ |
| GAT-sep | $88.75_{\pm 0.41}$ | $52.70_{\pm 0.62}$ | $93.91_{\pm 0.35}$ | $83.78_{\pm 0.43}$ | $76.79_{\pm 0.71}$ |
| GAT (LapPE) | $84.80_{\pm 0.46}$ | $44.90_{\pm 0.73}$ | $93.50_{\pm 0.54}$ | $84.99_{\pm 0.54}$ | $76.55_{\pm 0.84}$ |
| GAT (RWSE) | $86.62_{\pm 0.53}$ | $48.58_{\pm 0.41}$ | $92.53_{\pm 0.65}$ | $85.02_{\pm 0.67}$ | $77.83_{\pm 1.22}$ |
| GAT (DEG) | $85.51_{\pm 0.56}$ | $51.65_{\pm 0.60}$ | $93.04_{\pm 0.62}$ | $84.22_{\pm 0.81}$ | $77.10_{\pm 1.23}$ |
| Gated-GCN | $74.46_{\pm 0.54}$ | $43.00_{\pm 0.32}$ | $87.54_{\pm 1.22}$ | $77.31_{\pm 1.14}$ | $76.61_{\pm 1.13}$ |
| GCN | $73.69_{\pm 0.74}$ | $48.70_{\pm 0.63}$ | $89.75_{\pm 0.52}$ | $83.64_{\pm 0.67}$ | $76.09_{\pm 1.27}$ |
| GCN (LapPE) | $83.37_{\pm 0.55}$ | $44.35_{\pm 0.36}$ | $94.26_{\pm 0.49}$ | $84.95_{\pm 0.78}$ | $77.79_{\pm 1.34}$ |
| GCN (RWSE) | $84.84_{\pm 0.55}$ | $46.40_{\pm 0.55}$ | $93.84_{\pm 0.48}$ | $85.11_{\pm 0.77}$ | $77.81_{\pm 1.40}$ |
| GCN (DEG) | $84.21_{\pm 0.47}$ | $50.01_{\pm 0.69}$ | $94.14_{\pm 0.50}$ | $82.51_{\pm 0.83}$ | $76.96_{\pm 1.21}$ |
| CO-GNN$(\Sigma, \Sigma)$ | $91.57_{\pm 0.32}$ | $51.28_{\pm 0.56}$ | $95.09_{\pm 1.18}$ | $83.36_{\pm 0.89}$ | $80.02_{\pm 0.86}$ |
| CO-GNN$(\mu, \mu)$ | $91.37_{\pm 0.35}$ | $54.17_{\pm 0.37}$ | $97.31_{\pm 0.41}$ | $84.45_{\pm 1.17}$ | $76.54_{\pm 0.95}$ |
| SAGE | $85.74_{\pm 0.67}$ | $53.63_{\pm 0.39}$ | $93.51_{\pm 0.57}$ | $82.43_{\pm 0.44}$ | $76.44_{\pm 0.62}$ |
| **Graph Transformers** | | | | | |
| Exphormer | $89.03_{\pm 0.37}$ | $53.51_{\pm 0.46}$ | $90.74_{\pm 0.53}$ | $83.77_{\pm 0.78}$ | $73.94_{\pm 1.06}$ |
| NAGphormer | $74.34_{\pm 0.77}$ | $51.26_{\pm 0.72}$ | $84.19_{\pm 0.66}$ | $78.32_{\pm 0.95}$ | $68.17_{\pm 1.53}$ |
| GOAT | $71.59_{\pm 1.25}$ | $44.61_{\pm 0.50}$ | $81.09_{\pm 1.02}$ | $83.11_{\pm 1.04}$ | $75.76_{\pm 1.66}$ |
| GPS | $82.00_{\pm 0.61}$ | $53.10_{\pm 0.42}$ | $90.63_{\pm 0.67}$ | $83.71_{\pm 0.48}$ | $71.73_{\pm 1.47}$ |
| GPS$_{\text{GCN+Performer}}$ | $83.96_{\pm 0.53}$ | $48.20_{\pm 0.67}$ | $93.85_{\pm 0.41}$ | $84.72_{\pm 0.77}$ | $77.85_{\pm 1.25}$ |
| GPS$_{\text{GCN+Performer}}$ (RWSE) | $84.72_{\pm 0.65}$ | $48.08_{\pm 0.85}$ | $92.88_{\pm 0.50}$ | $84.81_{\pm 0.86}$ | $76.45_{\pm 1.51}$ |
| GPS$_{\text{GCN+Performer}}$ (DEG) | $83.38_{\pm 0.68}$ | $48.93_{\pm 0.47}$ | $93.60_{\pm 0.47}$ | $80.49_{\pm 0.97}$ | $74.24_{\pm 1.18}$ |
| GPS$_{\text{GAT+Performer}}$ (LapPE) | $85.93_{\pm 0.52}$ | $48.86_{\pm 0.38}$ | $92.62_{\pm 0.79}$ | $84.62_{\pm 0.54}$ | $76.71_{\pm 0.98}$ |
| GPS$_{\text{GAT+Performer}}$ (RWSE) | $87.04_{\pm 0.58}$ | $49.92_{\pm 0.68}$ | $91.08_{\pm 0.58}$ | $84.38_{\pm 0.91}$ | $77.14_{\pm 1.49}$ |
| GPS$_{\text{GAT+Performer}}$ | $85.54_{\pm 0.58}$ | $51.03_{\pm 0.60}$ | $91.52_{\pm 0.46}$ | $82.45_{\pm 0.89}$ | $76.51_{\pm 1.19}$ |
| GT | $86.51_{\pm 0.73}$ | $51.17_{\pm 0.66}$ | $91.85_{\pm 0.76}$ | $83.23_{\pm 0.64}$ | $77.95_{\pm 0.68}$ |
| GT-sep | $87.32_{\pm 0.39}$ | $52.18_{\pm 0.80}$ | $92.29_{\pm 0.47}$ | $82.52_{\pm 0.92}$ | $78.05_{\pm 0.93}$ |
| **Heterophily-Designated GNNs** | | | | | |
| CPGNN | $63.96_{\pm 0.62}$ | $39.79_{\pm 0.77}$ | $52.03_{\pm 5.46}$ | $73.36_{\pm 1.01}$ | $65.96_{\pm 1.95}$ |
| FAGCN | $65.22_{\pm 0.56}$ | $44.12_{\pm 0.30}$ | $88.17_{\pm 0.73}$ | $77.75_{\pm 1.05}$ | $77.24_{\pm 1.26}$ |
| FSGNN | $79.92_{\pm 0.56}$ | $52.74_{\pm 0.83}$ | $90.08_{\pm 0.70}$ | $82.76_{\pm 0.61}$ | $78.86_{\pm 0.92}$ |
| GBK-GNN | $74.57_{\pm 0.47}$ | $45.98_{\pm 0.71}$ | $90.85_{\pm 0.58}$ | $81.01_{\pm 0.67}$ | $74.47_{\pm 0.86}$ |
| GloGNN | $59.63_{\pm 0.69}$ | $36.89_{\pm 0.14}$ | $51.08_{\pm 1.23}$ | $73.39_{\pm 1.17}$ | $65.74_{\pm 1.19}$ |
| GPR-GNN | $64.85_{\pm 0.27}$ | $44.88_{\pm 0.34}$ | $86.24_{\pm 0.61}$ | $72.94_{\pm 0.97}$ | $55.48_{\pm 0.91}$ |
| H2GCN | $60.11_{\pm 0.52}$ | $36.47_{\pm 0.23}$ | $89.71_{\pm 0.31}$ | $73.35_{\pm 1.01}$ | $63.59_{\pm 1.46}$ |
| JacobiConv | $71.14_{\pm 0.42}$ | $43.55_{\pm 0.48}$ | $89.66_{\pm 0.40}$ | $68.66_{\pm 0.65}$ | $73.88_{\pm 1.16}$ |
| **Interleaved Multiscale (Ours)** | | | | | |
| IM-GCN | $83.53_{\pm 0.57}$ | $52.37_{\pm 0.66}$ | $91.80_{\pm 0.58}$ | $84.17_{\pm 0.71}$ | $78.17_{\pm 0.89}$ |
| IM-GatedGCN | $90.82_{\pm 0.59}$ | $54.01_{\pm 0.27}$ | $97.32_{\pm 0.83}$ | $85.10_{\pm 0.84}$ | $79.27_{\pm 0.91}$ |
| IM-GAT | $84.48_{\pm 0.32}$ | $51.16_{\pm 0.55}$ | $92.68_{\pm 0.49}$ | $85.21_{\pm 0.43}$ | $77.98_{\pm 1.01}$ |
| IM-FAGCN | $86.26_{\pm 0.44}$ | $52.81_{\pm 0.35}$ | $95.17_{\pm 0.84}$ | $84.49_{\pm 0.97}$ | $78.17_{\pm 1.06}$ |
| IM-GAT-sep | $89.93_{\pm 0.34}$ | $53.97_{\pm 0.58}$ | $96.15_{\pm 0.37}$ | $85.44_{\pm 0.40}$ | $77.92_{\pm 0.83}$ |
| IM-CO-GNN$(\Sigma, \Sigma)$ | $92.08_{\pm 0.33}$ | $53.11_{\pm 0.59}$ | $95.79_{\pm 0.96}$ | $85.25_{\pm 1.03}$ | $80.49_{\pm 0.92}$ |
| IM-CO-GNN$(\mu, \mu)$ | $92.00_{\pm 0.41}$ | $54.43_{\pm 0.41}$ | $97.39_{\pm 0.35}$ | $85.77_{\pm 1.05}$ | $78.92_{\pm 0.87}$ |

# D. Runtimes

We measure the runtimes of our IM-MPNN and compare it with baseline GNN backbone under different # scales settings and on three different datasets. The results are reported in Table 7, and show the effectiveness of IM-MPNN – it allows to achieve improved performance while maintaining a competitive computational demand in terms of runtimes, with a similar number of parameters.

Table 7: Training and inference runtime per epoch using an Nvidia RTX A6000 GPU.

| Method | scales | PascalVOC-SP | | COCO-SP | | Peptides-func | |
|---|---|---|---|---|---|---|---|
| | | params | time/epoch | params | time/epoch | params | sec/epoch |
| GCN | – | 490K | 8.00s | 500K | 78.95s | 486K | 2.00s |
| IM-GCN | 1 | 489K | 10.98s | 487K | 104.22s | 488K | 2.60s |
| IM-GCN | 2 | 482K | 14.22s | 495K | 132.64s | 495K | 3.41s |
| IM-GCN | 3 | 497K | 17.93s | 495K | 155.41s | 489K | 4.31s |
| IM-GCN | 4 | 472K | 21.86s | 489K | 183.94s | 478K | 5.16s |
| GINE | – | 450K | 8.03s | 409K | 76.18s | 491K | 1.90s |
| IM-GINE | 1 | 497K | 10.74s | 484K | 105.51s | 499K | 2.88s |
| IM-GINE | 2 | 496K | 14.04s | 484K | 130.33s | 479K | 3.85s |
| IM-GINE | 3 | 476K | 17.23s | 484K | 155.76s | 481K | 4.93s |
| IM-GINE | 4 | 491K | 20.83s | 499K | 182.76s | 481K | 6.02s |
| GatedGCN | – | 473K | 12.47s | 450K | 128.10s | 493K | 2.97s |
| IM-GatedGCN | 1 | 477K | 15.86s | 486K | 180.38s | 507K | 4.82s |
| IM-GatedGCN | 2 | 491K | 20.60s | 494K | 219.21s | 492K | 6.98s |
| IM-GatedGCN | 3 | 475K | 26.18s | 499K | 276.63s | 480K | 9.03s |
| IM-GatedGCN | 4 | 495K | 31.72s | 488K | 332.25s | 453K | 11.20s |

Table 8: Runtimes on the Questions dataset using 8-layer network with 256 channels on Nvidia RTX A6000 GPU.

| Method | milliseconds per epoch |
|---|---|
| GCN | 68.67 |
| CO-GNN | 211.43 |
| FAGCN | 104.85 |
| GatedGCN | 127.90 |
| GAT | 113.26 |
| GPS(Performer+GCN) | 412.07 |
| GPS(Transformer+GCN) | Out of memory |
| IM-GCN (Ours) | 151.74 |

# E. Additional Experiments

We've compared IM-MPNN results on PascalVOC-SP against two methods. The first is U-Net, which is a popular hierarchical method for node classification. The other is DRew, which is a graph rewiring method that aims to avoid over-squashing. Table 9 shows the results.

We also provide additional results on OGBN Arxiv. The results are reported in Table 10.

Table 9: Comparison with other methods.

| Method | PascalVOC-SP Test F1 ↑ |
|---|---|
| Graph U-Net | $0.1801_{\pm 0.0055}$ |
| DRew(GatedGCN) | $0.3909_{\pm 0.0051}$ |
| IM-GatedGCN | $\mathbf{0.4332}_{\pm 0.0045}$ |

Table 10: ORGB Arxiv results.

| Method | ORGB Arxiv Accuracy ↑ |
|---|---|
| GCN | $71.74_{\pm 0.29}$ |
| GAT | $71.95_{\pm 0.36}$ |
| IM-GCN | $\mathbf{73.89}_{\pm 0.21}$ |
| IM-GAT | $73.87_{\pm 0.16}$ |

