# OpenReview forum: "Improving the Effective Receptive Field of Message-Passing Neural Networks"
_ICML.cc/2025/Conference — ICML 2025 poster_

### Official Review · Reviewer_UcQT · 2025-03-07

**Overall Recommendation:** 2

**Summary:**

This paper introduces an architecture called Interleaved Multiscale Message-Passing Neural Networks (IM-MPNN) to address limitations in traditional Message-Passing Neural Networks (MPNNs), particularly the problem of over-squashing and the limited Effective Receptive Field (ERF). The key issue identified is that MPNNs struggle to capture long-range dependencies in graph-structured data due to an exponentially decaying influence of distant nodes, similar to the ERF limitations in Convolutional Neural Networks (CNNs).

## update after rebuttal
I keep my score, given the introduced addtional computation load, which may offset the performance gain. Also, it is not quite clear on how to choose the right scales for graphs.

**Claims And Evidence:**

The linear graph analysis (Section 3.1) assumes a simple topology, and the diffusion model (Section 3.2) relies on Laplacian-based assumptions. These may not fully generalize to complex, real-world graph structures. Table 6 shows increased runtimes with more scales. The claim of "maintaining computational efficiency" is supported by complexity analysis (O(|V| + |E|)), but practical runtime impacts are underexplored, especially for large-scale graphs.

**Essential References Not Discussed:**

Most relevant references are cited.

**Experimental Designs Or Analyses:**

The experimental designs are appropriate, including relevant datasets for evaluations. But more compasions with baselines on different benchmarks are needed.

**Methods And Evaluation Criteria:**

The IM-MPNN design is well-motivated, addressing over-squashing by expanding ERF through multiscale processing—a direct analogy to CNN solutions. The benchmarks (LRGB, heterophilic datasets, graph transfer) are appropriate.

**Other Comments Or Suggestions:**

1. Explain what different colors mean in Figure 1.
2. Figure 2, Line 67, "The the" -> "Then the"
3. Line 138, "nodes features the ℓ-th hidden layer" -> nodes features of the ℓ-th hidden layer

**Other Strengths And Weaknesses:**

Strengths:
1. The multiscale interleaving approach is a novel synthesis of CNN-inspired ERF enhancement, distinct from existing over-squashing solutions.
2. Improving ERF addresses a fundamental GNN limitation.
3. The proposed multiscale interleaved MP approach is well-presented and easy to follow.

Weaknesses:
1. It lacks comparisons with other approaches addressing the oversquashing problem, e.g., re-wiring.
2. Runtime increases with scales (Table 6), which suggest a trade-off not fully addressed, potentially limiting scalability for massive graphs.
3. The ERF analysis could explore more graph topologies or non-diffusion-based MPNNs to strengthen generality.

**Questions For Authors:**

1. Line 323-329, it is said to address the three questions in the experiments, and it is better to refer the question when discuss the corresponding results.
2. Table 1 and 2, it seems that the performance is not necessarily increasing with scales, can the authors further elaborate on this? How to choose the right scales for different graphs?
3. Can the authors compare the multiscale interleaved message passing with other approaches of addressing oversquashing, such as rewiring?

**Relation To Broader Scientific Literature:**

This paper mainly builds on the idea of increasing the ERF of CNN, and adapts it to graphs. Based on that, this work extends several key ideas in GNN research, i.e., oversquashing, Hierarchical GNNs, and Long-Range Dependencies.

**Theoretical Claims:**

There is no proof in this paper.

---

> ### Author Rebuttal · Authors · 2025-03-31
>
> We thank Reviewer UcQT for their thoughtful and constructive feedback, and for acknowledging our work being  *well-motivated*, *well-presented*, *distinct from existing solutions*, and *addressing a fundamental GNN limitation*. We are pleased to address your comments in detail below.
>
> 1. **Regarding comparison with rewiring:** During the rebuttal phase, we have experimented with a couple of methods that address oversquashing. The first is Graph U-Net, a hierarchical architecture that uses stages of pooling and unpooling to operate on different scales of the graph, with the main difference from IM-MPNN being that the U-Net approach only considers pairs of scales at a time. The second is DRew, a multi-hop method that utilizes rewiring, and we have chosen it because its implementation is compatible with the LRGB codebase, and because of its strong baseline performance. DRew was originally trained with the LRGB official hyperparameters, and not with the better-tuned ones presented in Toenshoff et al. (2023). Hence, for a fair comparison, we've rerun the experiments with those, changing only the width of the network to stay within the 500k parameter budget. The results for Graph U-Net, DRew, and our IM-GatedGCN are reported in the table below, further highlighting its effectiveness.
>
> | Method | PascalVOC-SP |
> |---|---|
> | Graph U-Net | 0.1801+-0.0055 |
> | DRew(GatedGCN) | 0.3909+-0.0051 |
> | IM-GatedGCN | 0.4332+-0.0045 |
>
>
> 2. **Regarding runtime:** Thank you for the comment.  We agree with the Reviewer that our IM-MPNN framework increases running times, as discussed in Section 4.3 reported in Appendix B (Table 6). However, Table 6 compares it to vanilla MPNNs, which are very fast and simple. It is expected that more complex architectures will require longer running times. For instance, on PascalVOC-SP, we measured a training runtime of 119.84 seconds using DRew-GatedGCN, while our IM-GatedGCN (with 4 levels) requires 31.72 seconds. That said, we agree that a leaner and faster method inspired by the interleaving multiscale principle is a promising direction for future work.
>
> 3. **Regarding the extension of the ERF analysis to more graph topologies:**
> The ERF analysis that we present in Section 3.2 relies on the connection between the combinatorial Laplacian being a discretization of the continuous Laplacian for some geometry. This connection is rather general and can hold for geometric problems characterized by graphs resembling an unstructured mesh discretizing a continuous domain. Figure 1, for example, presents such a mesh, discretizing a circular domain using triangulation. One can locate the nodes in space such that the combinatorial Laplacian (which is defined by the connectivity alone) will be equivalent to a finite element (FEM) discretization of a (minus) analytical Laplacian operator. Finite elements methods can discretize arbitrary domains in various dimensions, using quite a few types of elements and their mixes (triangular, rectangular, tetrahedral, etc.). Hence, since our analysis fits any FEM discretization (because it approximates a continuous equation in (10)), it is rather general, at least for geometric problems, and not limited only to a structured regular grid. Furthermore, the discussion in Section 4.1 regarding the scales will hold for any Finite Element discretization, as the discretization operators are scaled by the size of the elements (equivalent to $h^2$ in the paper). On the other hand, graphs are indeed more general than unstructured meshes, but we believe that our analysis gives intuition and motivation to use our method in such cases as well.
>
> 4. **Regarding lines 323-329 and addressing the three questions in the experiments:** While not explicitly answering these questions, we believe our experiments are constructed in a way that answers them. That is, Section 5.2 answers (i), and Sections 5.1 and 5.3 answer (ii) and (iii). We revised the text of the results section to reflect it more explicitly.
>
> 5. **Regarding typos and editorial suggestions:** Thank you for pointing those out. We have fixed our paper according to your guidance.
>
> ----
> We hope that you find our responses satisfactory, and that you will consider revising your score.

---

### Official Review · Reviewer_1PKF · 2025-03-13

**Overall Recommendation:** 3

**Summary:**

This work proposes a hierarchical coarsening method during GNN message passing in order to increase the effective receptive field while reducing over-squashing. The method is compared against datasets within the Long Range Graph Benchmark.

**Claims And Evidence:**

The experimental results well demonstrate the claims. Particularly with Figure 7, the IM-GCN shows strong performance on the given long-range dependence prediction task as the scale increases.

**Essential References Not Discussed:**

Most relevant references are discussed. One major work for hierarchical representations is DiffPool [1]. For hierarchical graph transformers, which utilize graph coarsening, another work to consider is ANS-GT [2].

[1] Ying, Zhitao, et al. "Hierarchical graph representation learning with differentiable pooling." Advances in neural information processing systems 31 (2018).

[2] Zhang, Zaixi, et al. "Hierarchical graph transformer with adaptive node sampling." Advances in Neural Information Processing Systems 35 (2022): 21171-21183.

**Experimental Designs Or Analyses:**

The experimental designs are standard for graph learning problems. Hierarchical MPNNs should be included in the experimental results, as these works are very closely related to the idea of graph coarsening.

**Methods And Evaluation Criteria:**

The methods and evaluation make sense for the problem.

**Other Comments Or Suggestions:**

See questions. I would increase my score with strong answers to my questions.

**Other Strengths And Weaknesses:**

The interleaved multiscale message passing is straightforward. Some extra discussion and comparison to hierarchical MPNNs would be important to highlight the differences to IM-MPNNs. As the paper is currently written, the novelty of this work is not entirely clear.

**Questions For Authors:**

(1) How is the pairing set P constructed?

(2) As mentioned previously, this work seems to resemble hierarchical MPNNs. What makes this work novel and sufficiently different from existing hierarchical MPNNs?

(3) What could the reason that IM-GatedGCN outperformed IM-GAT, despite the vanilla GAT outperforming the vanilla Gated-GCN? Why would interleaved multiscaling be more suited on a GatedGCN than a GAT?

(4) An interesting idea from a recent work is to perform what they call asynchronous aggregation during message passing [3]. This method is able to increase the effective receptive field as well by selectively aggregating messages from a node’s k-hop neighborhood and adding delays on when to aggregate each message. What do you see as the benefits of hierarchical coarsening over an approach like this?

[3] Jialong Chen, Tianchi Liao, Chuan Chen, and Zibin Zheng. 2024. Improving Message-Passing GNNs by Asynchronous Aggregation. In Proceedings of the 33rd ACM International Conference on Information and Knowledge Management (CIKM '24). Association for Computing Machinery, New York, NY, USA, 228–238. https://doi.org/10.1145/3627673.3679778

**Relation To Broader Scientific Literature:**

This work positions itself within the important discussion about the limitations of MPNNs, specifically about over-smoothing and over-squashing. The depth of graph models is limited by these factors. Therefore, works investigating ways to remedy these problems are important for the field.

**Theoretical Claims:**

The paper doesn’t put a large emphasis on theoretical findings.

---

> ### Author Rebuttal · Authors · 2025-03-31
>
> We thank Reviewer 1PKF for their thoughtful and constructive feedback. We are happy to read that you found our *claims well demonstrated* and that the *paper tackles a problem important for the field*, and we are pleased to address your comments in detail below.
>
> 1. **Regarding related hierarchical MPNNs (DiffPool and ANS-GT):** Thank you for pointing out these related works. DiffPool is a coarsening (pooling) method for graphs. In contrast, our proposed IM-MPNN utilizes coarsening operations as a building block, and its main contribution is in the way in which the hierarchical representations of graph features are processed. Thus, it is possible to use DiffPool within IM-MPNN as its coarsening operation instead of Graclus. We chose to work with Graclus thanks to its popularity. We have clarified this important distinction in our paper.
> ANS-GT is a variation of graph transformers that uses a preprocessed graph coarsening within its attention mechanism. It differs from IM-MPNN in two ways: First, our IM-MPNN uses coarsening as part of a message-passing network. Second, it interleaves multiple coarsening levels instead of a single one. In our revised paper, we now added this discussion and citations to DiffPool and ANS-GT. Thank you.
>
> 2. **Regarding the difference from other hierarchical MPNNs:** To the best of our knowledge, this is the first MPNN-based method to use multiple graph scales during the entire processing stage of the neural network, which is different from U-Net based approaches that only consider pairs of scales at a time. We demonstrate that it outperforms the baselines on datasets that require long-range interactions, and inspired by your comment, we also include an empirical comparison to highlight the effectiveness of IM-MPNN compared with Graph U-Net, in the table below. We have included this discussion in our revised paper.
>
> | Method | PascalVOC-SP|
> |---|---|
> | Graph U-Net | 0.1801+-0.0055 |
> | IM-GatedGCN | 0.4332+-0.0045 |
>
> 3. **Regarding the pairing set P:** The pairing is done using the Graclus clustering, as discussed in our response to point 1. However, it is not limited to this specific method and can be obtained with other clustering methods as well. Following your question and a request from reviewer EHuP, we added more information about Graclus to the appendix.
>
> 4. **Reasoning on IM-GatedGCN vs. IM-GAT results:**
> We agree with the reviewer that this is an interesting topic. However, we would like to point out that this is a phenomenon that can happen empirically. For example, in Table 5, we see that GPS+Performer+GCN outperforms GPS+Performer+GAT on three of the five datasets. Therefore, while this is interesting to explore, investigating it is out of the scope of this paper.
>
> 5. **Regarding asynchronous aggregation during message passing:** We thank the reviewer for bringing the work to our attention. The method, aAsyn, suggests a multi-hop aggregation in an asynchronous approach. We did not find the code for this method in order to compare results with it. Hence, we compared with DRew (Gutteridge et al., 2023), which is another multi-hop approach, as discussed in response to Reviewer UcQT, to show the benefits of IM-MPNN over it.  We agree that aAsyn is relevant to our work, and we now cite and discuss it in our revised paper.
>
> | Method | PascalVOC-SP |
> |---|---|
> | DRew(GatedGCN) | 0.3909+-0.0051 |
> | IM-GatedGCN | 0.4332+-0.0045 |
>
> ----
> We hope that you find our responses satisfactory, and that you will consider revising your score.

---

> > ### Comment · Reviewer_1PKF · 2025-04-04
> >
> > Thank you to the authors for their thoughtful response. I believe the extra discussion and comparison to other related baselines strengthens this work. I will update my score appropriately.

---

> > > ### Author Response · Authors · 2025-04-08
> > >
> > > We thank you for your positive feedback and for adjusting your score. We appreciate your thoughtful review and are happy you found the additional discussion and comparisons to clarify our work and strengthen it.
> > > Sincerely, Authors

---

### Official Review · Reviewer_EHuP · 2025-03-14

**Overall Recommendation:** 4

**Summary:**

The paper addresses the challenges faced in capturing long-range interactions in GNNs due to limited effective receptive field of the message passing mechanism and proposes a novel architecture based on a hierarchical coarsening of graph to improve communication between distant nodes.

## update after rebuttal:
Following the clarifications and additional results provided by the authors during the rebuttal, I believe the complementary nature of the proposed approach could be valuable to the GNN community. Therefore, I have raised my score.

**Claims And Evidence:**

- The main claim that the contribution of one node to another node's output decaying exponentially by the distance between them is shown and analyzed on different synthetic graph types.

- While the results in Table 3 show  IM-MPNN to achieve the best performance, some more baselines reported in Table 5 seem to be more competitive, particularly CO-GNN. Although I understand all results can not be reported in the main paper due to space limitation, I believe comparison with most recent competitors such as CO-GNNs should be included in the main results table.

**Essential References Not Discussed:**

None that I can recall.

**Experimental Designs Or Analyses:**

The experiments and analyses are reasonably well designed and conducted.

**Methods And Evaluation Criteria:**

- The proposed technique is analyzed on different graph types and evaluated on relevant real-world benchmark datasets.

-  IM-MPNN, while interesting, is more time consuming in practice although its time complexity  is a linear factor of a regular MPNN. Since the predictive performance is competitve or only a small improvement over a baseline such as CO-GNNs , it would also be nice to see how IM-MPNN compares empirically to CO-GNNs (and/or other such competing methods in addition to basic MPNNs such as GCN already shown in Table 6) in terms of runtime.

**Other Comments Or Suggestions:**

A brief basic explanation of the method used to select nodes pairs for graph coarsening could be helpful for readers not familiar with Graclus algorithm.

**Other Strengths And Weaknesses:**

**Strengths**: The paper provides an interesting perspective to an imporant problem in the GNN landscape,  is well-written with illustrations aiding comprehension.

**Weaknesses**: Runtime for even 4 scale levels in hierarchical coarsening is realtively quite high. An empirical comparison of runtime between IM-MPNN and competitive baselines with similar performance such  as CO-GNNs and evaluation on large-scale OGB datasets could be a valuable addition.

**Questions For Authors:**

1. Would the number of scales (and the number of nodes in each scale level) be a hyperparameter to be tuned? Is there any guideline for its selection depending on some properties of the input graph? Does performance tend to level out or decrease if the number of scales are further increased?

2. How many layers did the IM-MPGNNs generally constitute of?

3. Can IM-MPNN use heterophily designated GNNs or others as the underlying MPNN? For example, IM-GATsep or IM-FAGCN? If so, how would they be expected to perform? Is IM-MPNN complimentary with any GNN backbone or is there a case where a certain type of GNN could be deterimental to IM-MPNN?

4. Since the performance of IM-MPNNs and CO-GNNs is similar, do  IM-MPNNs hold any other advantages over CO-GNNs, such as efficiency?

**Relation To Broader Scientific Literature:**

The paper offers a relatively new perspective on effectively enabling long range interactions whereas most existing solutions are based on re-wiring strategies or specifically designed architectures and thus adds value to the current GNN literature.

**Theoretical Claims:**

The theoretical claims seem correct but the mathematical details were not checked in detail.

---

> ### Author Rebuttal · Authors · 2025-03-31
>
> We thank Reviewer EHuP for the thoughtful and constructive feedback and for finding our paper *interesting*, *well-written*, valuable to the current GNN literature, and offering a *different perspective on enabling long-range interactions*. We are pleased to address your comments in detail below.
>
> All the results and discussions were added to our revised paper, and we think they improved our paper, thank you.
>
> 1. **Regarding organizing Table 5:** Our goal was to compare with multiple relevant methods, hence Appendix Table 5 includes 33 methods. Following your suggestion, CO-GNN is now included in Table 3 (main paper), as well as our IM-COGNN -- please see details below.
> 2. **Re runtimes:**  We agree with that IM-MPNN increases runtime, as discussed in Section 4.3 and Appendix B (Table 6). However, IM-MPNN retains the asymptotic complexity of its backbone MPNN, typically linear in the number of nodes and edges, and Tables 1, 2, and 6 show that IM-MPNN significantly improves performance over its backbone. We also kindly note that methods like DRew (Gutteridge et al., 2023) requires higher runtimes. For example, on PascalVOC-SP, DRew-GatedGCN takes 119.84s vs. 31.72s for IM-GatedGCN (4 levels). Following your advice, we now report runtimes on the Questions dataset using 8-layer networks with 256 channels on an Nvidia A6000. The results demonstrate that our method achieves strong performance while keeping runtime comparable to methods like CO-GNN.
>
> |Method|milliseconds per epoch|
> |---|---|
> |GCN|68.67|
> |CO-GNN|211.43|
> |FAGCN|104.85|
> |GatedGCN|127.90|
> |GAT|113.26|
> |GPS(Performer+GCN)|412.07|
> |GPS(Transformer+GCN)|Out of memory|
> |IM-GCN (Ours)|151.74|
>
> 3. **Regarding comparison with CO-GNN:** We incorporated IM-MPNN into CO-GNN and found it further improves CO-GNN’s strong baseline, as shown in the table below.
>
> |Method|Roman.|Amazon.|Mine.|Tolo.|Ques.|
> |---|---|---|---|---|---|
> |CO-GNN($\Sigma$,$\Sigma$)|91.57+-0.32|51.28+-0.56|95.09+-1.18|83.36+-0.89|80.02+-0.86|
> |CO-GNN($\mu$,$\mu$)|91.37+-0.35|54.17+-0.37|97.31+-0.41|84.45+-1.17|76.54+-0.95|
> |IM-CO-GNN($\Sigma$,$\Sigma$) (Ours)|92.08+-0.33|53.11+-0.59|95.79+-0.96|85.25+-1.03|80.49+-0.92|
> |IM-CO-GNN($\mu$,$\mu$) (Ours)|92.00+-0.41|54.43+-0.41|97.39+-0.35|85.77+-1.05|78.92+-0.87|
>
> 4. **Regarding OGB dataset:** During the rebuttal, we ran experiments with a couple of IM-MPNN options on OGBN ARXIV. The results are reported in the table below, and they suggest that our approach is beneficial also for larger graphs.
>
> |Method|OGBN Arxiv (Acc)|
> |---|---|
> |GCN|71.74+-0.29|
> |GAT|71.95+-0.36|
> |IM-GCN (Ours)|73.89+-0.21|
> |IM-GAT (Ours)|73.87+-0.16|
>
> 5. **Regarding the Graclus algorithm:** Our focus was not on the coarsening method, as IM-MPNN can benefit from other approaches in the literature. However, we agree that an explanation is helpful and have added it to the revised paper. Thank you.
>
> 6. **Regarding number of scales:** The Reviewer is correct that the number of scales is a hyperparameter. Like width and depth in neural networks, performance eventually plateaus or drops, which can be due to several factors. First, to stay within a parameter budget, we reduce network width when increasing scales, which may limit performance. Second, adding more scales may offer no benefit once the interaction range is sufficient for the data. For example, on a graph with a diameter of 16 and 3 coarsening levels, a node at the coarsest scale may already span most of the graph, making further scaling unnecessary. Thus, the optimal number of scales depends on the graph’s diameter.
>
> 7. **Regarding the number of layers:** We use IM-MPNN to enhance MPNN backbones and therefore follow the hyperparameters from prior work. For example, in IM-GatedGCN on PascalVOC-SP, we use the GatedGCN settings from Toenshoff et al. (2023), including 10 layers.
>
> 8. **Regarding IM-MPNN with heterophily designated GNNs:** To address your insightful question, we provide results of IM-GATsep and IM-FAGCN on heterophilic datasets below. These show that IM-MPNN can enhance various MPNN backbones and improve their performance.
>
> |Method|Roman.|Amazon.|Mine.|Tolo.|Ques.|
> |---|---|---|---|---|---|
> |GAT-sep|88.75+-0.41|52.70+-0.62|93.91+-0.35|83.78+-0.43|76.79+-0.71|
> |FAGCN|65.22+-0.56|44.12+-0.30|88.17+-0.73|77.75+-1.05|77.24+-1.26|
> |IM-GAT-sep (Ours)|89.93+-0.34|53.97+-0.58|96.15+-0.37|85.44+-0.40|77.92+-0.92|
> |IM-FAGCN (Ours)|86.26+-0.44|52.81+-0.35|95.17+-0.84|84.49+-0.97|78.17+-1.06|
>
> 9. **Regarding IM-MPNNs and CO-GNNs:** CO-GNN is a strong method that explores learning MPNN actions. We see IM-MPNN as a complementary contribution, offering interleaved hierarchical processing that can be combined with CO-GNN. To demonstrate this, we now include IM-COGNN in the table above, showing that IM-MPNN can further enhance CO-GNN’s strong baseline.
>
> ---------
> We hope that you find our responses satisfactory, and that you will consider revising your score.

---

> > ### Comment · Reviewer_EHuP · 2025-04-08
> >
> > I thank the authors for the clarifications and appreciate their effort in providing further experiments. The complementary nature of their proposed approach could be helpful to various types of GNN architectures for tackling different tasks. Therefore, I have raised my score.

---

> > > ### Author Response · Authors · 2025-04-09
> > >
> > > Thank you for your feedback and for raising your score. We are glad that our response helped clarify the complementary nature of our approach for different GNN architectures and that the additional experiments addressed your concerns.
> > > Sincerely, Authors.

---

### Official Review · Reviewer_U4Ms · 2025-03-14

**Overall Recommendation:** 4

**Summary:**

This paper proposes a new messages passing strategy to expand the receptive field of GNNs by transmitting information between graphs at multiple scales, and theoretical analysizes the influence decay of message passing  along the path between nodes. The experiments are conducted on long-range graph benchmarks to validate the effectiveness of multiscale message interaction.

## update after rebuttal:

My concerns in the first review phase have been addressed by the authors with empirical justification, so I raised my score.

**Claims And Evidence:**

Yes.

**Essential References Not Discussed:**

No.

**Experimental Designs Or Analyses:**

Yes. The experimental designs are sound, which follow the widely used protocals.

**Methods And Evaluation Criteria:**

Yes.

**Other Comments Or Suggestions:**

1. Messages are constrained to pass through the coarsen graphs in the adjacent layers, why not passing messages between any pair of coarsened graphs (including the original one)? what is the benefit of the current choice?
2. It is somehow counterintuitive to adopt  pairwise coarsening according to the topology of the heterophilic graphs, as there is a good chance for connected nodes in that setting that have different features/labels, but grouping them into one node as well as feature aggregation according to eq.(17) may lead to invalid node features. How can we explain the outperformance of IM-GNN in heterophilic graphs?

**Other Strengths And Weaknesses:**

Strengths:
1. The paper is well-written and easy to follow.
2. It is a good idea to establish short-cut message passing channels between long-range nodes via coarsened graphs.
3. The message passing interleaved among different scaled graphs  is meaningful, leading to  impressive results.

Weaknesses: Why the proposed method is superior to other methods that are also able to capture the long-range dependence on heterophilic graphs is not discussed.

**Questions For Authors:**

In Eq.(20), what are X_1 and X_2? Can it be formulated in the product of coarsening matrix and feature matrix, which I guess will be more neat.

**Relation To Broader Scientific Literature:**

The key contribution of this work is to build message passing among multi-layered graphs, namely, original graph and its coarsened graphs at different scale, which shares idea with (Yang et al. "SeBot: Structural Entropy Guided Multi-View Contrastive
Learning for Social Bot Detection", in KDD'24), where the hierarchy of the graph in question is constructed with some coarsening method like structure entropy, but the messages are only allowed to transfer from higher level to low level (directionally).

**Theoretical Claims:**

checked.

---

> ### Author Rebuttal · Authors · 2025-03-31
>
> We thank Reviewer U4Ms for their thoughtful and constructive feedback. We appreciate you finding our paper *well-written and easy to follow*, and that the proposed method is  *meaningful* with *impressive results*, and we are pleased to address your comments in detail below.
>
>
> 1. **Regarding SeBot (Yang et al. 2024)**: Thank you for the reference. The hierarchical information processing in SeBot appears to be based on a Graph U-Net structure, focusing on how to perform coarsening between scales. In contrast, our IM-MPNN uses existing coarsening operations, and its contribution is in the unique multiscale interleaving approach to enhance the ERF of MPNNs, as reflected by the results in the Table below, where our method significantly outperforms Graph U-Net. It might be possible (and interesting to try) to use SeBot with IM-MPNN in future works. We added this discussion and a citation to SeBot in our revised paper.
>
> | Method | PascalVOC-SP |
> |---|---|
> | Graph U-Net | 0.1801$_{\pm 0.0055}$ |
> | IM-GatedGCN | 0.4332$_{\pm 0.0045}$ |
>
> 2. **Regarding other methods on heterophilic graphs:** Thank you for the suggestion. As noted by Reviewer EHuP, our IM-MPNN offers a different approach compared with existing methods, that rely mostly on rewiring or designated architectures, while our IM-MPNN  offers a hierarchical approach to allow the propagation of information from distant nodes. Moreover, IM-MPNN can be combined with these methods, to further improve performance, as shown in our results provided in the response to Reviewer EHuP. We added the discussion and results to the revised paper.
>
> 3. **Regarding messages between any pair of coarsened graphs:** We appreciate the interesting question. We have tried similar ideas of passing information between different scales of the graph, which also adds to the computational complexity of the architecture. However, we did not see an improvement over the proposed interleaving approach.
>
> 4. **Regarding the coarsening of heterophilic graphs:** Thank you for the insightful question.  We would like to kindly note that an IM-MPNN layer processes the original resolution features (as well as other scales) on their own, followed by their aggregation. Thus, in our IM-MPNN, we only add information to the original resolution node features, which are aggregated from distant nodes, using the Graclus pooling algorithm. We agree with the Reviewer that the specific choice of pooling and aggregation may be improved for heterophilic graphs, and it will be interesting to explore such approaches in future works. Finally, we note that our experiments consistently indicate that our IM-MPNN improves downstream performance, also in heterophilic graphs, including when combined heterophily-designated MPNNs, as shown in our added experiments in our response to Reviewer EHuP. We added this fruitful discussion to our revised paper. Thank you.
>
>
> 5. **Regarding Eq. (20):** Thank you for pointing out our typo, we have now corrected the equation according to your guidance, and it reads (apologies for the line separation. There were issues with Openreview's rendering of the equation):
>
> $$x_{q=(i,j)}^{(s,\ell+1)} = \tilde{x}_{q=(i,j)}^{(s,\ell)}$$
>
> $$+ W_{l2h}^{(s,\ell)}\frac{1}{2}(\tilde{x}_i^{(s-1,\ell)}+\tilde{x}_j^{(s-1,\ell)})$$
>
> $$+ W_{h2l}^{(s,\ell)}\tilde{x}_{(q,p)}^{(s+1,\ell)}.$$
>
> ----------
> We hope that you find our responses satisfactory, and that you will consider revising your score.

---

### Decision · Program_Chairs · 2025-05-01

**Decision:**

Accept (poster)

**Comment:**

This work introduces a formal definition of "effective receptive field" in GNNs, inspired by the literature on Convolutional Neural Networks. The authors present a novel architecture based on hierarchical coarsening and scale mixing to extend the receptive field of GNNs. The proposed method is evaluated empirically on challenging benchmarks such as the Long-Range Graph Benchmark, showing that it outperforms existing approaches in modeling long-range dependencies and alleviating the over-squashing problem.

The paper was reviewed by four reviewers. All four initially raised concerns, mainly about novelty, certain design choices, applicability to heterophilic graphs, and computational overhead. These concerns were addressed satisfactorily in detailed rebuttals. Specifically, the authors clarified differences from prior work and provided additional experiments, including runtime benchmarks and performance results on Open Graph Benchmark datasets and heterophilic graph settings, showing comparable or better performance without sacrificing efficiency.

I share their view that this is a well-written and clearly structured paper presenting an interesting method supported by solid evidence. I believe this is a significant contribution to the area of graph learning and therefore recommend acceptance.

For the revised version, I suggest that the authors include the additional results and discussion provided during the rebuttal phase, especially the details of the strategy for selecting the number of scales.